# Multiscale guidance of protein structure prediction with heterogeneous cryo-EM data

**Rishwanth Raghu**
Princeton University
rraghu@princeton.edu

**Axel Levy**
Stanford University
axlevy@stanford.edu

**Gordon Wetzstein**
Stanford University
gordon.wetzstein@stanford.edu

**Ellen D. Zhong**
Princeton University
zhonge@princeton.edu

## Abstract

Protein structure prediction models are now capable of generating accurate 3D structural hypotheses from sequence alone. However, they routinely fail to capture the conformational diversity of dynamic biomolecular complexes, often requiring heuristic MSA subsampling approaches for generating alternative states. In parallel, cryo-electron microscopy (cryo-EM) has emerged as a powerful tool for imaging near-native structural heterogeneity, but is challenged by arduous pipelines to transform raw experimental data into atomic models. Here, we bridge the gap between these modalities, combining cryo-EM density maps with the rich sequence and biophysical priors learned by protein structure prediction models. Our method, CryoBoltz, guides the sampling trajectory of a pretrained biomolecular structure prediction model using both global and local structural constraints derived from density maps, driving predictions towards conformational states consistent with the experimental data. We demonstrate that this flexible yet powerful inference-time approach allows us to build atomic models into heterogeneous cryo-EM maps across a variety of dynamic biomolecular systems including transporters and antibodies.

## 1 Introduction

Proteins and other macromolecules in our cells are constantly vibrating, deforming, and interacting with other surrounding molecules. Characterizing the variability of their atomic structures, i.e., of the relative 3-dimensional (3D) locations of their atoms, can deepen our understanding of the complex chemical mechanisms underlying basic biological systems. For example, understanding how a driver mutation can alter the probability of certain conformational states has applications ranging from drug design [25] to molecular engineering [21].

A variety of experimental methods for protein structure determination have been developed, with X-ray crystallography and cryo-electron microscopy (cryo-EM) being the most widely used today. In cryo-EM, a series of breakthroughs in both hardware [80, 9, 24, 29] and software [69, 27, 68, 64] led to the so-called "resolution revolution" [47], resulting in routine near-atomic resolution structure determination for well-behaved purified protein samples. Cryo-EM, in particular, also possesses the ability to measure and reconstruct the conformational landscape of dynamic biomolecular complexes [68, 64, 94, 10, 62, 63]. However, these experiments are still complex, costly, and time consuming, requiring expensive microscopes and facilities, as well as hours to days of data processing through iterative computational pipelines [49]. Notably, current reconstruction algorithms only output 3D "density maps" that approximate the electron scattering potential of the molecule.

39th Conference on Neural Information Processing Systems (NeurIPS 2025).

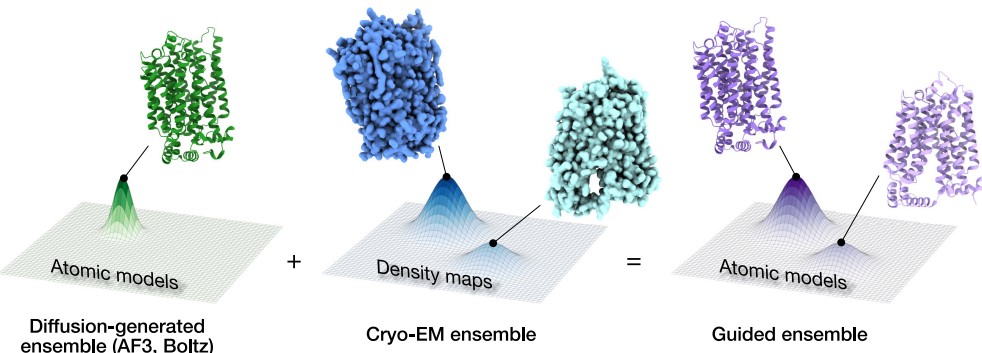

Figure 1: **Problem statement.** Diffusion-based structure prediction models [1, 88] can sample different conformations, but the generated ensemble is generally peaked around a single conformation. A cryo-EM experiment probes the full conformational landscape but current reconstruction algorithms only provide density maps, not atomic models. With CryoBoltz, we guide the diffusion process in atomic space with experimental cryo-EM measurements to increase sample diversity and more faithfully reflect the true conformational ensemble.

Fitting atomic models within these maps is a significant computational challenge for which existing methods [85, 51, 82, 81, 35, 39] only provide partial solutions that need to be manually refined.

Building on decades of data acquisition, processing, and curation [7, 8], machine-learning-based sequence-to-structure models were developed and trained on publicly available structural data [41, 3, 52, 1]. These models, however, are still trained to map a given sequence to a unique, most likely structure and are therefore bound to viewing proteins as static objects. Shifting structure prediction models to a dynamic paradigm constitutes one of today's main challenges for structural biology.

Recent exploratory lines of work have attempted to address this outstanding challenge. MSA subsampling methods, for example, rely on randomly masking input sequence data to broaden the diversity of output structures [87, 23, 43, 36]. Despite results showing improved diversity on specific systems, MSA subsampling methods remain an active area of research, with no clear consensus yet regarding their performance. Moreover, these methods are not well suited for complexes that can adopt many different conformational states or a continuum of conformational states. Other works, including AlphaFlow [40] and BioEmu [50], investigated incorporating physics-based molecular dynamics simulation as additional training data. These works also showed greater variability among output structures but were mainly demonstrated on small peptides, additionally requiring costly training and relying on simulations that may not capture realistic atomic motions.

Here we introduce a method, CryoBoltz, that leverages heterogeneous cryo-EM data to guide the sampling process of a diffusion-based structure prediction algorithm (Figure 1). Our implementation is based on Boltz-1 [88], an open-source sequence-conditioned diffusion model heavily inspired by the state-of-the-art model AlphaFold3 [1]. Through a multiscale guidance mechanism, CryoBoltz combines the structural information learned by the pretrained diffusion model with experimentally-captured data, producing structures consistent with the cryo-EM data. Importantly, our method does not require an additional training step, while effectively mitigating the single-structure bias of current structure prediction models. We demonstrate results on both synthetic and real cryo-EM maps of dynamic biomolecular complexes.

## 2 Background

### 2.1 Diffusion-Based Sampling in AlphaFold3

Recent advances in protein structure prediction from sequences are exemplified by major breakthroughs such as AlphaFold2 [41] and AlphaFold3 [1]. While AlphaFold2 predicts static structures with remarkable accuracy, AlphaFold3 introduces a diffusion modeling head within its structure module, enabling generative sampling of different conformations, conditioned on the same sequence.

Specifically, AlphaFold3 utilizes a diffusion model operating directly in the space of atomic coordinates, $\mathbf{x} \in \mathbb{R}^{N \times 3}$ where $N$ represents the number of modeled atoms. Given a sequence $\mathbf{s}$, we call $p_0(\mathbf{x}|\mathbf{s})$ the distribution of conformations of the folded protein (or complex) at ambient temperature. We then call $p_t(\mathbf{x}|\mathbf{s})$ the marginal distribution of conformations obtained by sampling $\mathbf{x}_0$ from $p_0(\mathbf{x}|\mathbf{s})$ and simulating the *forward* diffusion process

$$\mathrm{d}\mathbf{x} = \mathbf{f}(\mathbf{x}, t)\mathrm{d}t + g(t)\mathrm{d}\mathbf{w}, \tag{1}$$

from 0 to $t$, where $\mathbf{f}(\mathbf{x}, t)$ and $g(t)$ are predefined drift and diffusion functions while $\mathrm{d}\mathbf{w}$ represents a standard Wiener process in atomic coordinate space. The drift and diffusion functions are chosen such that $p_T(\mathbf{x}|\mathbf{s}) \approx \mathcal{N}(\mathbf{0}, \mathbf{I})$ for some $T \in \mathbb{R}$. One way to sample from the target distribution $p_0(\mathbf{x}|\mathbf{s})$ is then to sample $\mathbf{x}_T$ from $\mathcal{N}(\mathbf{0}, \mathbf{I})$ and simulate the *reverse* diffusion process

$$\mathrm{d}\mathbf{x} = \left(\mathbf{f}(\mathbf{x}, t) - g(t)^2 \nabla_{\mathbf{x}} \log p_t(\mathbf{x}|\mathbf{s})\right) \mathrm{d}t + g(t)\mathrm{d}\mathbf{w} \tag{2}$$

from $T$ to 0 [2, 34]. In the above equation, the score function $\nabla_{\mathbf{x}} \log p_t(\mathbf{x}|\mathbf{s})$ is unknown and implicitly depends on the target distribution. AlphaFold3 therefore uses an approximation of the score function, called $s_\theta(\mathbf{x}, \mathbf{s}, t)$. This "score model" can be obtained using a finite set of samples from $p_0(\mathbf{x}|\mathbf{s})$ and a training strategy based on denoising score matching [84, 76].

Boltz-1 closely follows the architecture and framework of AlphaFold3 with minor modifications, achieving comparable accuracy in predicting biomolecular complex structures [88].

## 2.2 Likelihood-Based Guidance

In an "inverse problem", one aims at recovering an unknown object $\mathbf{x}$ from a measurement $\mathbf{y}$, given a known "likelihood model" $p(\mathbf{y}|\mathbf{x})$. Inverse problems are often framed as posterior sampling problems, i.e., they aim at sampling from the posterior $p(\mathbf{x}|\mathbf{y})$. Using Bayes' rule, the posterior can be decomposed as a product of the likelihood, $p(\mathbf{y}|\mathbf{x})$, and the prior distribution over $\mathbf{x}$.

In this context, several works have recently shown that pretrained diffusion models can be interpreted as implicitly defined priors and therefore used to solve posterior sampling problems [38, 75, 11, 12, 74, 45, 46, 86], as surveyed in [20]. Effectively, these methods are able to "guide" a diffusion model using a measurement $\mathbf{y}$ and its corresponding likelihood model. The key insight of these works lies in noticing that the score function of the posterior can be re-written as a sum: $\nabla_{\mathbf{x}} \log p_t(\mathbf{x}|\mathbf{y}) = \nabla_{\mathbf{x}} \log p_t(\mathbf{x}) + \nabla_{\mathbf{x}} \log p(\mathbf{y}|\mathbf{x}_t = \mathbf{x})$. The first term is directly approximated by the pretrained score model $s_\theta(\mathbf{x}, \mathbf{s}, t)$, but the challenge lies in the second term. The conditional probability $p(\mathbf{y}|\mathbf{x}_t)$ can be written as a conditional expectation $\mathbb{E}_{\mathbf{x}_0 \sim p(\mathbf{x}_0|\mathbf{x}_t)}[p(\mathbf{y}|\mathbf{x}_0)]$, but approximating this expectation with Monte Carlo samples is not a computationally tractable option, because sampling $n$ times from the conditional distribution $p(\mathbf{x}_0|\mathbf{x}_t)$ requires solving $n$ differential equations. In ScoreALD, Jalal et al. [38] first suggested to replace the latter distribution with a Dirac delta centered on $\mathbf{x}$, effectively replacing $p(\mathbf{y}|\mathbf{x}_t)$ with $p(\mathbf{y}|\mathbf{x})$. Despite promising results on low-noise and linear inverse problems, ScoreALD tends to drive samples off the diffusion manifold, i.e., in regions where $p_t(\mathbf{x}) \ll 1$, where the score model was only sparsely supervised and is therefore highly inaccurate. To mitigate this issue, Chung et al. [12] suggested in the DPS algorithm to center the Dirac delta distribution on $\hat{\mathbf{x}}_\theta(\mathbf{x}, t) = \mathbb{E}_{\mathbf{x}_0 \sim p(\mathbf{x}_0|\mathbf{x}_t = \mathbf{x})}[\mathbf{x}_0]$, which can be expressed as an affine function of $s_\theta(\mathbf{x}, t)$ with Tweedie's formula [77, 66, 26]. The *guided* reverse diffusion process is therefore defined as

$$\mathrm{d}\mathbf{x} = \left(\mathbf{f}(\mathbf{x}, t) - g(t)^2 s_\theta(\mathbf{x}, t) - \lambda(t) \underbrace{\nabla_{\mathbf{x}} \log p(\mathbf{y}|\mathbf{x}_0 = \hat{\mathbf{x}}_\theta(\mathbf{x}, t))}_{\tilde{s}_\theta(\mathbf{y}, \mathbf{x}, t)}\right) \mathrm{d}t + g(t)\mathrm{d}\mathbf{w}, \tag{3}$$

where $\tilde{s}_\theta(\mathbf{y}, \mathbf{x}, t)$ is an additional guidance term.

In parallel to the guidance-based approach, other works have attempted to frame the posterior sampling problem as a problem of variational inference [30, 55], but remain limited by the expressivity of the variational family (e.g., Gaussian distributions [55]) or by the necessity to repeatedly solve initial-value problems [30]. Most recently, MCMC-based strategies like DAPS [91] proposed to correct previous sampling methods with an equilibration step based on MCMC sampling (e.g., Langevin dynamics or Hamiltonian Monte Carlo) and showed improved performance on high-noise or highly nonlinear problems. However, owing to the simplicity and effectiveness of the DPS algorithm for our problem of interest, we base our guiding mechanism on the DPS framework.

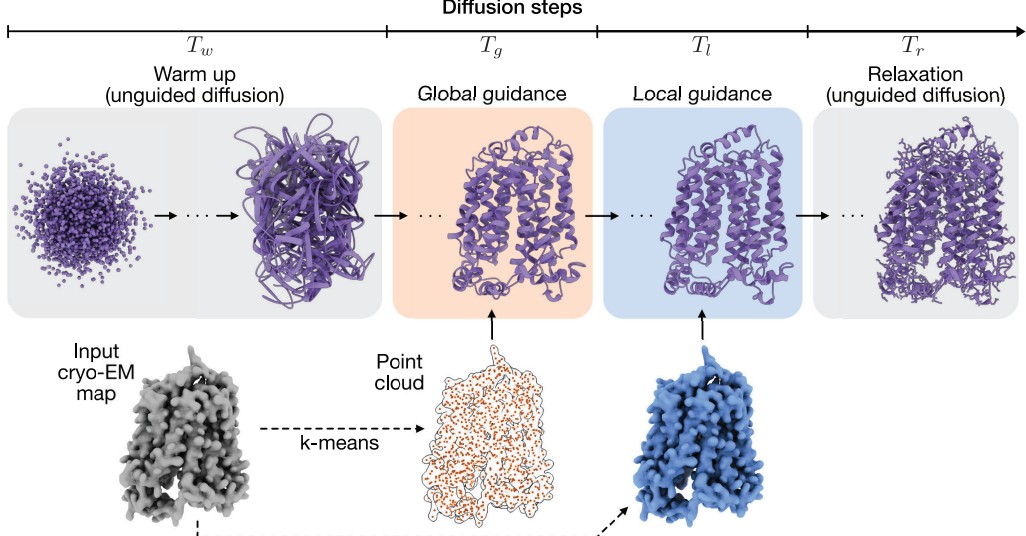

Figure 2: **Overview of the guidance mechanism.** The diffusion process starts with a "warm up" stage where the score model is only conditioned on the sequence. During "global guidance", the structure is guided to minimize its distance to a point cloud representation of the input cryo-EM map. During "local guidance", it is guided to maximize its consistency with the original input map. Finally, the last "relaxation" steps are unguided and allow the model to correct fine-grained details.

## 2.3 Forward Model for Cryo-EM Maps

Cryo-EM reconstructs a 3D electron scattering potential from many independent 2D projection images of frozen biomolecular complexes. The reconstructed density map $\mathbf{y}$ is usually represented as a 3D array: $\mathbf{y} \in \mathbb{R}^{w \times h \times d}$. In the forward model, a given structure's cryo-EM density is typically modeled as a sum of Gaussian form factors centered on each atom. Formally, the observed density map can be modeled as $\mathbf{y} = \mathcal{B}(\Gamma(\mathbf{x}, \mathbf{s})) + \eta$ [22], where $\Gamma$ is an operator that sums isotropic Gaussians centered on each heavy (non-hydrogen) atom in $\mathbf{x}$. Their amplitudes and variances are tabulated [33] and typically depend on the chemical elements in $\mathbf{x}$. $\mathcal{B}$ represents the effect of "B-factors" [44] and can be viewed as a spatially dependent blurring kernel modeling molecular motions and/or signal damping by the transfer function of the electron microscope. Finally, $\eta$ models i.i.d. Gaussian noise.

## 3 Methods

In this section, we describe how CryoBoltz uses an input cryo-EM map to guide a diffusion process in atomic space. Our implementation is based on the DPS algorithm (Section 2.2) [12].

### 3.1 Overview: Multiscale Guidance

Because the forward model turning atomic models $\mathbf{x}$ into density maps is highly nonlinear (Section 2.3), the likelihood function $p(\mathbf{y}|\mathbf{x})$ is multimodal w.r.t. $\mathbf{x}$, making the posterior $p(\mathbf{x}|\mathbf{y}) \propto p(\mathbf{y}|\mathbf{x})p(\mathbf{x})$ rugged and hard to sample from with score-based methods. In order to regularize the target ensemble distribution in the early diffusion steps, CryoBoltz uses a global-to-local guidance strategy that ignores high-resolution information until the later diffusion steps.

First, we use $T_w$ steps of unguided reverse diffusion, only conditioning the score model on the sequence $\mathbf{s}$, to bootstrap (i.e., "warm up") the atomic model and obtain a structure close to the one the diffusion model is initially biased towards (Figure 1, left), following Equation 2. We then use

$T_g$ steps of a "global guidance" strategy, further described in Section 3.2, followed by $T_l$ steps of a "local guidance" stage, described in Section 3.3. Finally, the last $T_r$ diffusion steps are unguided to help solve high-resolution inconsistencies (e.g., steric clashes). Figure 2 provides an overview of our multiscale guidance strategy.

## 3.2 Global Guidance

At the beginning of the global guidance stage, the density map $\mathbf{y} \in \mathbb{R}^{w \times h \times d}$ is transformed into a 3D point cloud $\mathbf{Y} \in \mathbb{R}^{k \times 3}$. This conversion is done using the weighted $k$-means clustering algorithm with a predefined number of clusters $k$ dependent on the number of atoms in the system and the voxel size of the map (see details in the Appendix). This point cloud, inspired by the volumetric shape constraints in Chroma [37], provides a compact and low-resolution representation of the map that can be used to efficiently guide the diffusion process towards the global shape of the protein complex. A key benefit of the point cloud representation is that the distance between $\mathbf{x}$ and $\mathbf{Y}$ can be defined using standard distances derived from optimal-transport theory, like the Sinkhorn divergence $\mathfrak{D}(\mathbf{x}, \mathbf{Y})$, a regularized version of the Wasserstein distance [60]. In the first global guidance step, the intermediate sample $\mathbf{x}$ is aligned with the density map prior to computing the Sinkhorn divergence (see Appendix for more details).

Following the DPS framework, we define the guided diffusion process using Equation 3 (with sequence-conditioning in the pretrained score model), where the guidance term is defined as

$$\tilde{s}_\theta(\mathbf{y}, \mathbf{x}, \mathbf{s}, t) = -\nabla_{\mathbf{x}} \mathfrak{D}(\hat{\mathbf{x}}_\theta(\mathbf{x}, \mathbf{s}, t), \mathbf{Y}). \tag{4}$$

The schedule of the guidance strength $\lambda(t)$ is described in the Appendix. Note here that the global guidance term is not directly derived from a physics-based likelihood model, but rather defined heuristically in order for the atomic model $\mathbf{x}$ to fit the low-resolution details of the cryo-EM map.

## 3.3 Local Guidance

During local guidance, the original density map is used and the guidance term of Equation 3 is directly derived from the forward model described in Section 2.3, i.e.,

$$\tilde{s}_\theta(\mathbf{y}, \mathbf{x}, \mathbf{s}, t) = -\nabla_{\mathbf{x}} \|\mathbf{y} - \mathcal{B}(\Gamma(\hat{\mathbf{x}}_\theta(\mathbf{x}, \mathbf{s}, t), \mathbf{s}))\|^2. \tag{5}$$

At this stage, the guidance term includes all the structural information captured in the density map, including higher resolution details, and derives directly from physics-based assumptions on the cryo-EM forward model and noise model.

## 3.4 Related Work

Our work proposes to guide a pretrained diffusion model operating on atomic coordinates using experimental cryo-EM data. Equivalently, our method can be seen as a model building method leveraging a pretrained diffusion model as a regularizer.

First developed for X-ray crystallography [13], model building methods were later adapted to operate on cryo-EM data [85, 51, 82, 81] but the obtained atomic models were often incomplete and needed refinement [73]. Machine-learning-based methods were also developed, either relying on U-Net architectures [72, 92, 61] or combining 3D transformers with Hidden Markov Models [32]. He et al. [35] first made use of sequence information in EMBuild, and ModelAngelo [39] has recently established a new state of the art for automated *de novo* model building. Combining a GNN-based architecture with preprocessed sequence information [65], ModelAngelo outperforms previous approaches. However, its performance relies on high-resolution maps (below 4 Å) and often yields incomplete models on blurry, low-resolution data (Figure 5, for example). As a result, manual model building remains the prevailing solution in these challenging regimes, particularly in those involving flexible or heterogeneous complexes.

The possibility of using structure prediction models as regularizers for 3D reconstruction problems was only demonstrated very recently. In ROCKET, Fadini et al. [28] introduced a method to use AlphaFold2 [41] as a prior for building atomic models that are consistent with cryo-EM, cryo-ET or X-ray crystallography data. The method regularizes the problem by transferring the optimization from atomic space to the latent space of AlphaFold2. In contrast, our method leverages AlphaFold3's

Table 1: **Quantitative evaluation with synthetic maps (STP10 [6] and CH67 antibody [70]) and ablation study.** We report the Root Mean Square Deviation (RMSD) for all atoms, the C$\alpha$ RMSD, and the template-modeling (TM) score. For CH67, we also report the RMSD for the C$\alpha$ atoms in the CDR H3 loop (local RMSD). The last two columns show an ablation study on the guidance mechanism. The mean across 3 replicates is reported for the best of 25 samples (lowest all-atom RMSD). Random MSA subsampling of Boltz-1 [88] (Boltz-1 + MSA sub.) is run with MSA depths of 64, 128, 256, 512, and 1024, each producing 5 samples. **Bold** indicates best value. AF3 is AlphaFold3 [1].

| *Structure* | *Metrics* | CryoBoltz | Boltz-1 | Boltz-1 + MSA sub. | AF3 | Local only | Global only |
|---|---|---|---|---|---|---|---|
| STP10 (inward) | RMSD (Å, ↓) | **1.057** | 3.815 | 3.768 | 1.263 | 3.860 | 1.287 |
| | C$\alpha$ RMSD (Å, ↓) | **0.371** | 3.554 | 3.513 | 0.623 | 3.559 | 0.778 |
| | TM score (↑) | **0.998** | 0.863 | 0.865 | 0.994 | 0.862 | 0.990 |
| STP10 (outward) | RMSD | **0.888** | 2.656 | 2.542 | 4.478 | 2.722 | 1.164 |
| | C$\alpha$ RMSD | **0.440** | 2.419 | 2.295 | 4.228 | 2.458 | 0.779 |
| | TM score | **0.997** | 0.948 | 0.953 | 0.828 | 0.946 | 0.991 |
| CH67 antibody | RMSD | **1.048** | 1.961 | 1.954 | 1.887 | 1.443 | 1.281 |
| | C$\alpha$ RMSD | **0.637** | 1.469 | 1.522 | 1.453 | 0.945 | 0.880 |
| | Local RMSD | **1.269** | 3.120 | 3.270 | 3.191 | 1.718 | 1.899 |
| | TM score | **0.994** | 0.972 | 0.969 | 0.971 | 0.990 | 0.988 |

diffusion-based structure module for efficient optimization directly in atomic space. Other works investigated the possibility to guide diffusion-based models using experimental data [54, 53] but were only used to process X-ray crystallography data. In this modality, each measurement provides an average of the contribution of each conformation in the crystal, which inherently limits the extent to which structural variability can be analyzed. Finally, ADP-3D [48] demonstrated diffusion-based model refinement using cryo-EM measurements, but the method requires an initial model (provided, for example, by ModelAngelo [39]) and was not compared to existing structure prediction methods.

# 4 Results

## 4.1 Experimental Setup

**Datasets and metrics.** We evaluate our method on six biomolecular systems. For two of them, we guide CryoBoltz with synthetic density maps (STP10 [6] and CH67 antibody [70]) and use real, experimental maps for the other four systems (P-glycoprotein [14], Pma1, CYP102A1, and YbbAP [56]). We chose these four systems because (1) they have two or more density maps corresponding to different conformational states; (2) the corresponding atomic models were deposited after the Boltz-1 [88] training cutoff; (3) they are composed of two or fewer unique chains, so that an accurate unguided Boltz-1 prediction can be obtained; and (4) the sequence is shorter than 2,200 residues, which we found to be the max sequence length that could fit in Boltz-1. For three of these systems (P-glycoprotein, CYP102A1, and YbbAP), at least one map is of lower resolution (>4 Å). To assess the quality of a generated structure, we align it to the deposited (reference) structure pairing $\alpha$-carbons, then compute C$\alpha$ root mean square deviation (RMSD), all-atom RSMD, and template-modeling (TM) score [93]. We sample 25 structures for each of three model replicates. We additionally report map-model fit metrics in Supplementary Table B2.

**Baselines.** We compare CryoBoltz against the diffusion-based structure prediction models Boltz-1 [88] and AlphaFold3 [1]. We additionally evaluate Boltz-1 with MSA subsampling, producing 5 samples each for 64, 128, 256, 512, and 1024 randomly drawn MSA sequences. On the experimental datasets, we also compare our results to those obtained with the model building algorithm ModelAngelo [39].

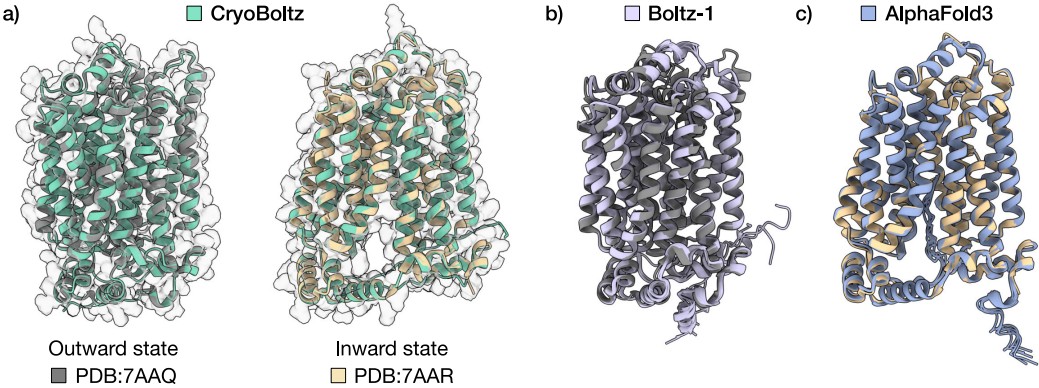

**Figure 3:** **Results on STP10 [6]. a)** CryoBoltz can predict both the outward and inward states, when guided with the respective cryo-EM density map. Best sample (lowest all-atom RMSD across all model replicates) is shown. **b, c)** Current structure prediction models are biased towards one of these states. Boltz-1 [88] only predicts the outward state **(b)** while AlphaFold3 [1] only predicts the inward state **(c)**. Five best samples relative to the outward and inward PDB structures, respectively, are shown.

## 4.2   Synthetic datasets

**STP10.** We demonstrate our method on the sugar transporter protein STP10, a plant protein that switches between inward-facing and outward-facing conformations as it shuttles substrates across the cell membrane [6]. From the deposited atomic models of these structures (PDB:7AAQ, 7AAR) [4, 5], we generate synthetic density maps at a resolution of 2 Å using the *molmap* function in ChimeraX [59]. In Figure 3, we show that density-guided diffusion allows for accurate modeling of both conformational states. While unguided Boltz-1 only samples the outward conformation, CryoBoltz guidance drives the rearrangement of helices to sample the inward conformation. MSA subsampling slightly improves the accuracy of Boltz-1 but still only samples the outward conformation. AlphaFold3, in contrast, only samples the inward conformation. As seen in Table 1, CryoBoltz not only models both conformations, but also improves the accuracy of the predictions over their unguided counterparts, achieving an all-atom RMSD below 1 Å for the outward state.

**CH67 antibody.** CH67 is an antibody whose Fab domain binds the influenza hemagglutinin receptor during the human immune response [70]. Responsible for this interaction is the complimentarity-determining region (CDR) H3, a short loop that is highly variable across antibody families and thus is modeled poorly by protein structure prediction methods. To assess the ability of our method to cope with the lower resolutions typically obtained for antibody cryo-EM maps, we simulate a 4 Å density map of the CH67 Fab domain (PDB:4HKX) [71]. In Figure 4, we show that CryoBoltz accurately models the CDR H3 loop, correctly placing the backbone and most of the side chains. With some samples achieving a local RMSD below 1 Å on this region (Supplementary Figure B2), our method improves over Boltz-1 and AlphaFold3 (Table 1), due to both better global modeling of the full Fab structure as well as local modeling of the H3 loop itself (Supplementary Figure B1).

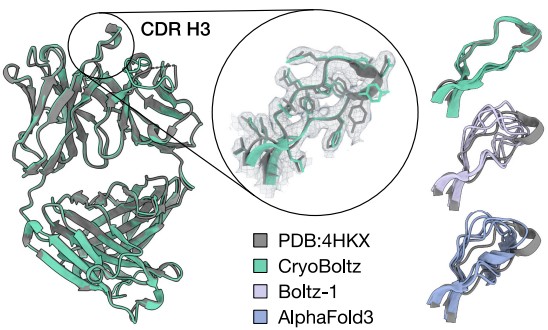

**Figure 4:** **Results on CH67 antibody [70].** CryoBoltz fits the CDR H3 loop more accurately than Boltz-1 [88] and AlphaFold3 [1]. Three best samples (lowest all-atom RMSD across all model replicates) are shown on the right.

Table 2: **Quantitative evaluation using real cryo-EM density maps of four biomolecular systems.** We report the mean all-atom RMSD (Å), Cα RMSD (Å) and TM score for the best of 25 samples (lowest all-atom RMSD) across 3 model replicates. We indicate the resolution of the input map (res.) as well as the completeness (comp.) of the model built by ModelAngelo (MA) [39] (percentage of modeled over deposited residues). Pgp is P-glycoprotein [14] and CYP is CYP102A1. **Bold** indicates best value.

| | | CryoBoltz | | | Boltz-1 | | | AlphaFold3 | | | ModelAngelo | |
|---|---|---|---|---|---|---|---|---|---|---|---|---|
| *Structure* | Res. (Å) | RMSD all | RMSD Cα | TM score | RMSD all | RMSD Cα | TM score | RMSD all | RMSD Cα | TM score | Comp. (%) | TM score |
| Pgp (apo) | 4.3 | **1.382** | **1.208** | **0.989** | 6.994 | 7.194 | 0.767 | 3.827 | 3.865 | 0.904 | 40.3 | 0.361 |
| Pgp (inward) | 4.4 | **1.348** | **1.187** | **0.989** | 5.630 | 5.692 | 0.828 | 2.692 | 2.663 | 0.947 | 18.3 | 0.134 |
| Pgp (occluded) | 4.1 | **1.727** | **1.677** | **0.979** | 2.929 | 2.904 | 0.942 | 3.440 | 3.420 | 0.921 | 2.3 | 0.010 |
| Pgp (collapsed) | 4.4 | **1.309** | **1.261** | **0.988** | 3.425 | 3.412 | 0.917 | 4.568 | 4.554 | 0.864 | 2.5 | 0.010 |
| Pma1 (active) | 3.25 | **2.046** | **1.776** | **0.973** | 2.987 | 2.752 | 0.935 | 6.628 | 6.389 | 0.769 | 91.5 | 0.889 |
| Pma1 (inhibited) | 3.52 | **1.999** | **1.590** | **0.979** | 6.140 | 5.829 | 0.794 | 8.017 | 7.776 | 0.723 | 72.8 | 0.721 |
| CYP (open) | 6.5 | **4.167** | **3.946** | **0.957** | 8.532 | 8.439 | 0.788 | 6.490 | 6.361 | 0.890 | 0.0 | 0.000 |
| CYP (closed) | 4.4 | **2.004** | **1.552** | **0.990** | 8.784 | 8.667 | 0.743 | 3.585 | 3.391 | 0.946 | 18.9 | 0.102 |
| YbbAP (bound) | 3.66 | **1.320** | **0.678** | **0.997** | 3.623 | 3.339 | 0.928 | 3.749 | 3.480 | 0.922 | 81.7 | 0.801 |
| YbbAP (unbound) | 4.05 | **2.454** | **2.039** | **0.974** | 7.842 | 7.654 | 0.776 | 4.022 | 3.744 | 0.913 | 55.4 | 0.548 |

## 4.3 Experimental datasets

**P-glycoprotein.** P-glycoprotein is a membrane transporter that conducts cellular export of toxic compounds including chemotherapy drugs, making it an important therapeutic target for inhibition [14]. We test our method on experimental density maps corresponding to four states in the transport cycle: the apo state, inward state, occluded state, and collapsed state (EMD-40226, 40259, 40258, 40227). We mask the maps around their corresponding deposited atomic models (PDB:8GMG, 8SA1, 8SA0, 8GMJ) [15, 18, 17, 16] in order to remove non-protein detergent density, which is a byproduct of sample preparation for transmembrane proteins. As shown in Figure 5 and Table 2, CryoBoltz samples the full set of conformations, outperforming baselines in all metrics across all four states. We additionally find that ModelAngelo predictions are highly incomplete, with only between 2.3% and 40.3% of residues modeled. We provide ModelAngelo with the original maps as they lead to marginally higher performance.

**Pma1.** We use two experimental maps of a Pma1 monomer (EMD-64135, 64136), corresponding to the active and inhibited states (PDB:9UGB, 9UGC) [89, 90] of this ATPase. Unguided Boltz only samples the active state, whereas guidance also samples the inhibited state with an all-atom RMSD of 2.00 Å and Cα RMSD of 1.59 Å (Table 2, Figure 6). AlphaFold3 predictions are not accurate with respect to either state.

**CYP102A1.** We use two experimental maps of CYP102A1 (EMD-27534, 27536), of resolutions 4.4 Å and 6.5 Å, corresponding to the open and closed states (PDB:8DME, 8DMG) [78, 79] of this oxygenase. Guidance improves the all-atom RMSD of the closed state from 8.78 Å to 2.00 Å over unguided Boltz-1, whereas for the lower resolution open state map, a more modest improvement from 8.53 Å to 4.17 Å is observed. ModelAngelo only models 18.9% of the 4.4 Å map and none of the 6.5 Å map.

**YbbAP.** We use two experimental maps of the YbbAP transporter [56] (EMD-51292, 51291), corresponding to states in which ATP is bound or unbound (PDB:9GE7, 9GE6) [58, 57]. Unguided Boltz only samples the bound state, which guidance further improves to an all-atom RMSD of 1.32 Å. The unbound state is additionally obtained through guidance. We observe that MSA subsampling also allows Boltz-1 to sample the unbound sample in some model replicates (Supplementary Table B1).

## 4.4 Ablations

To validate our multiscale approach, we ablate the model by exclusively running the global guidance phase or local guidance phase. As shown in Table 1, while global guidance alone often improves metrics over baselines, local guidance further boosts accuracy by fitting higher-resolution details.

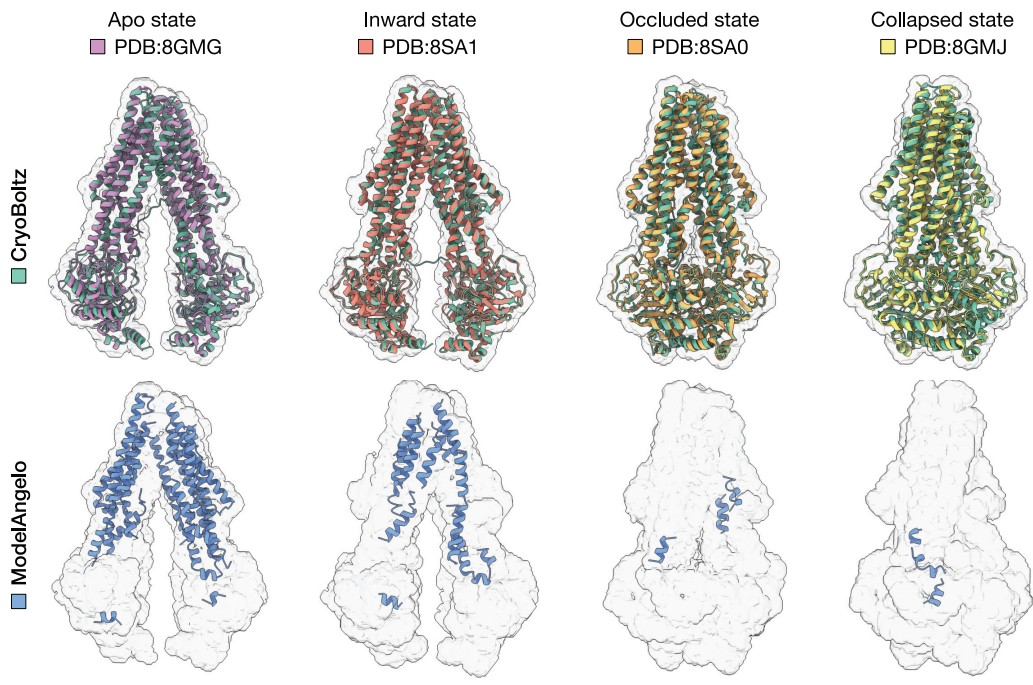

Figure 5: **Results on P-glycoprotein [14].** CryoBoltz recovers distinct states of the protein using four experimental cryo-EM maps. ModelAngelo [39] produces highly incomplete models.

Local guidance alone also improves performance over unguided Boltz-1, but is ultimately insufficient in driving large conformational changes. We report ablations for real density maps in Supplementary Table B1. The combination of local and global guidance leads to better accuracy than either one alone for most of the maps.

# 5   Discussion

This work introduces a guidance mechanism that increases the capability of current diffusion-based structure prediction models. The guiding information, derived from experimental cryo-EM measurements, biases sampling towards atomic models consistent with the observed data. Our method does not require any retraining or finetuning and can be used on top of any available model, thereby making it possible to benefit from their continuous improvement. Through experiments on both synthetic and experimental data, we show that CryoBoltz can increase the diversity of sampled conformations – revealing states that are missed by existing diffusion models – and predict more accurately the structure of regions that are key to function, like CDR loops in antibodies. On experimental data, we show that state-of-the-art model building methods can fail and CryoBoltz, leveraging knowledge acquired from large-scale datasets of protein structures, can fit atomic models within minutes, saving hours of manual refinement. With the increasing availability of predicted structures as priors for cryo-EM model building, the principled validation of the resulting atomic models, especially those derived from lower resolution maps, remains an open question.

An important limitation of CryoBoltz is the limited stability of optimization, due to the multimodality of the likelihood $p(\mathbf{y}|\mathbf{x})$. This instability is mitigated by sampling several structures simultaneously and selecting the best fit a posteriori, but this comes at the cost of increased memory and time consumption. Importantly, CryoBoltz also relies on the base (unguided) model being able to provide a good initialization during the "warm-up" stage, which we found not to be the case on several systems (e.g., DSL1/SNARE complex [19], full IgG antibody [67]). Future directions for this work therefore include exploring ways to stabilize guidance, mitigating the drift towards "off-manifold" regions (where the diffusion model is highly inaccurate), or getting rid of heuristic choices like the specific duration of each guidance stage. Finally, when having access to $N$ cryo-EM maps, associated with

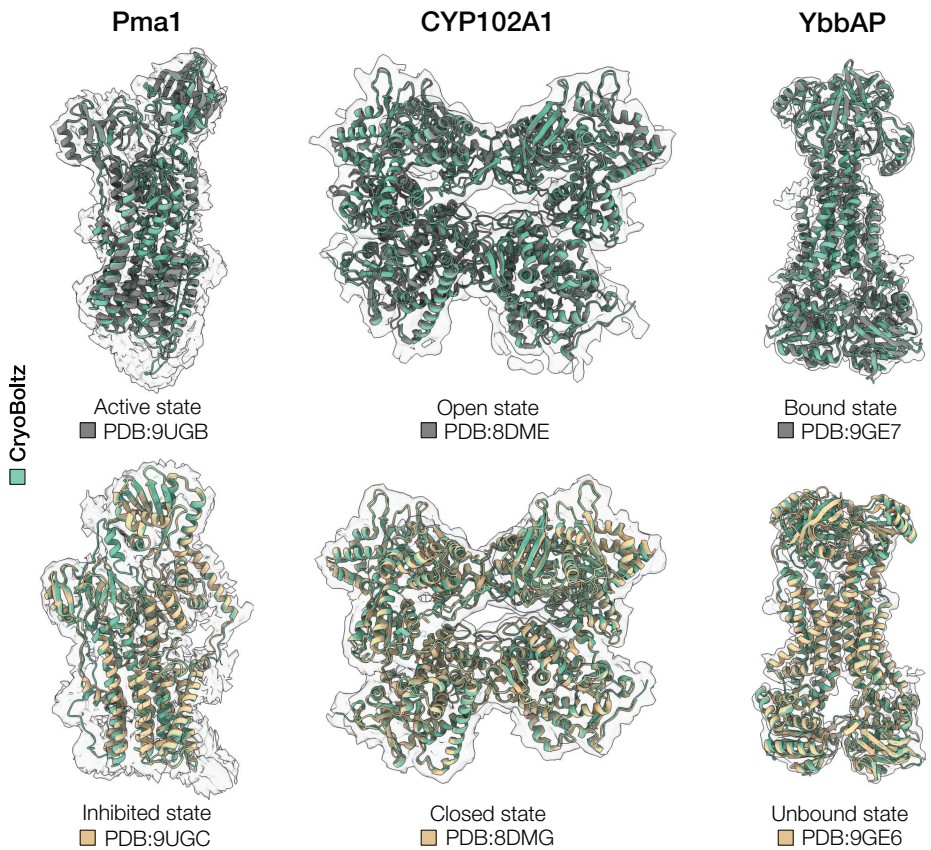

Figure 6: **Results on Pma1, CYP102A1, and YbbAP [56].** Cryoboltz guides the atomic model into two distinct conformations per complex based on experimental density maps.

similar conformations, exploring the possibility to optimize a unique deformation model instead of $N$ independent models constitutes an interesting avenue for future work.

**Conclusion.** In this study, we demonstrate the possibility of increasing the sample diversity of state-of-the-art generative models using experimental cryo-EM data. Doing so, we hope to contribute to the ongoing community effort towards efficiently exploring the conformational landscape of macromolecules with machine learning models.

## Acknowledgements

The authors acknowledge the use of computing resources at Princeton Research Computing, a consortium of groups led by the Princeton Institute for Computational Science and Engineering (PICSciE) and Office of Information Technology's Research Computing. The Zhong lab is grateful for support from the Princeton Catalysis Initiative, Princeton School of Engineering and Applied Sciences, Chan Zuckerberg Imaging Institute, Janssen Pharmaceuticals, and Generate Biomedicines. The funders had no role in study design, data collection and analysis, decision to publish or preparation of the manuscript.

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

# Appendix

## A    Supplemental Methods

### A.1    Point cloud construction

For global guidance, the input density map is converted to a point cloud by performing weighted $k$-means clustering on the voxel coordinates, where the weights are given by the map intensity values. $k$ is set to $\lfloor N/(4r^3)\rfloor$, where $N$ is the number of atoms in the system and $r$ is the voxel size of the map in Å. Prior to clustering, density values below a threshold may be set to 0, and connected density components below a size threshold may be removed ("dusted"). See Experimental Details for dataset-dependent values, where applicable.

### A.2    Simulated map construction

For local guidance, a density map is simulated from the sample using a single Gaussian per atom whose amplitude is given by atomic number, as implemented in the default *molmap* function of ChimeraX [59]. The map is simulated to match the voxel size and nominal resolution of the input map.

### A.3    Alignment to density map

For efficient optimization of the point cloud guidance term, the intermediate sample must be aligned to the input density map. We obtain an unguided sample and dock it into the map using the ChimeraX *fitmap* function [59]. Prior to the first step of guidance, the intermediate sample is aligned to the unguided sample via the Kabsch algorithm [42].

### A.4    Sinkhorn divergence

For two point clouds $X \in \mathbb{R}^{N \times 3}$ and $Y \in \mathbb{R}^{M \times 3}$, the entropy-regularized optimal transport distance is given by

$$\text{OT}_\epsilon(X, Y) = \min_{\gamma \in \mathbb{R}_+^{N \times M}} \sum_i^N \sum_j^M \gamma_{ij} C_{ij} - \epsilon \sum_i^N \sum_j^M \gamma_{ij} \log \gamma_{ij}$$

where $\gamma$ is a *transport plan* whose rows each sum to $1/N$ and columns each sum to $1/M$. The first term is the Wasserstein distance between the point clouds, where the cost $C_{ij}$ between two points is the squared Euclidean distance $\frac{1}{2}||X_i - Y_j||_2^2$. The second term is the entropy of $\gamma$, which allows for tractable and differentiable optimization and is controlled by the regularization strength $\epsilon$. As the entropy term introduces an approximation error to the true Wasserstein distance, the Sinkhorn divergence corrects for this and is defined as

$$\mathfrak{D}(X, Y) = \text{OT}_\epsilon(X, Y) - \frac{1}{2}\text{OT}_\epsilon(X, X) - \frac{1}{2}\text{OT}_\epsilon(Y, Y)$$

We use the GeomLoss library for efficient optimization of the objective $\mathfrak{D}$ with respect to the sample coordinates [31]. The *reach* parameter is set to 10 while all others are set to their defaults.

### A.5    Guidance schedule

For the synthetic datasets, the numbers of steps in the guidance phases are $T_w = 125$, $T_g = 25$, $T_l = 25$, and $T_r = 25$. For the experimental datasets, the numbers of steps are $T_w = 100$, $T_g = 50$, $T_l = 25$, and $T_r = 25$. During the global guidance phase (for all datasets), the guidance strength is annealed along a cosine schedule from 0.25 to 0.05, i.e., $\lambda(t) = 0.05 + \frac{1}{2}(0.25 - 0.05)(1 + \cos(\frac{\pi t}{T_g}))$. During the local guidance phase, the guidance strength is made constant at $\lambda(t) = 0.5$.

### A.6    Experimental Details

**STP10.** Deposited structures of the inward and outward conformations (PDB: `7AAQ`, `7AAR`) [4, 5] were stripped of non-protein entities. Synthetic density maps of 2 Å resolution and 1 Å voxel size were

generated using the ChimeraX *molmap* function [59], then padded to dimension $w = h = d = 100$. The wild-type protein sequence corresponding to `PDB:7AAQ` was given as input.

**CH67 Antibody.** The deposited structure of the antibody Fab (`PDB:4HKX`) was stripped of non-protein entities and the bound hemagglutinin receptor. A synthetic density map of 4 Å resolution and 1 Å voxel size was generated and padded to dimension $w = h = d = 100$. For CryoBoltz and Boltz-1 [88], the *step_scale* parameter, which controls the temperature of the sampling distribution, is set at 3.0 to increase CDR H3 loop accuracy.

**P-glycoprotein.** The experimental maps of four conformational states (`EMD-40226, 40259, 40258, 40227`) were masked around their corresponding deposited models (`PDB:8GMG, 8SA1, 8SA0, 8GMJ`) [15–18] to remove micelle density, using the ChimeraX *volume zone* function [59]. A padding of 5 voxels was then added to each side.

**Pma1.** The experimental maps of two conformational states (`EMD-64135, 64136`) were cropped to a tight box at a density threshold of 0.35 then padded by 10 voxels on each side. This removed empty background regions for computational efficiency. During the global guidance phase of the method, the maps were thresholded at a value of 0.35.

**CYP102A1.** The experimental maps of two conformational states (`EMD-27534, 27536`) were cropped to a tight box at a density threshold of 1.25 then padded by 10 voxels on each side. During the global guidance phase of the method, the maps were thresholded at a value of 1.25.

**YbbAP.** The experimental maps of two conformational states (`EMD-51292, 51291`) were cropped to a tight box at a density threshold of 0.005 then padded by 10 voxels on each side. During the global guidance phase of the method, the maps were thresholded at a value of 0.005, and dusted with size threshold 100.

## A.7 Computational resources

All experiments were performed on a single Nvidia A100 GPU with 80 GB VRAM.

# B Supplemental Results

## B.1 Map-model fit and statistical significance

In Table B2, we report evaluation metrics and confidence bounds on the best generated sample as assessed by map-model fit. Three replicates of each method are run to produce 25 samples per replicate. Samples are ranked according to the real-space correlation coefficient (RSCC), which is the Pearson correlation between the input density map and a map simulated from the sample [83]. For the unguided baselines, we first align each sample to the deposited model via the Kabsch algorithm [42]. CryoBoltz demonstrates statistically significant improvement over baselines across a majority of maps and metrics.

## B.2 Spread of sample quality

In Figures B2 and B3, we visualize the full distribution of RMSD values over all samples produced by CryoBoltz, Boltz-1 and AlphaFold3. For nearly all maps, a majority of CryoBoltz samples are more accurate than those produced by the baselines.

Table B1: **Additional baselines and ablations for experimental cryo-EM density maps.** Extending Table 2, we report accuracy metrics for MSA subsampling of Boltz-1, as well as ablations of the local and global guidance phases. We report the mean all-atom RMSD (Å), Cα RMSD (Å) and TM score for the best of 25 samples (lowest all-atom RMSD) across 3 model replicates. Random MSA subsampling of Boltz-1 (Boltz-1 + MSA sub.) is run with MSA depths of 64, 128, 256, 512, and 1024, each producing 5 samples. Pgp is P-glycoprotein and CYP is CYP201A1. **Bold** indicates best value(s).

| | CryoBoltz | | | Boltz-1 + MSA subsampling | | | Local only | | | Global only | | |
|---|---|---|---|---|---|---|---|---|---|---|---|---|
| | RMSD | RMSD | TM | RMSD | RMSD | TM | RMSD | RMSD | TM | RMSD | RMSD | TM |
| *Structure* | all | Cα | score | all | Cα | score | all | Cα | score | all | Cα | score |
| Pgp (apo) | **1.382** | **1.208** | **0.989** | 6.902 | 7.099 | 0.770 | 5.897 | 6.026 | 0.810 | 1.501 | 1.335 | 0.986 |
| Pgp (inward) | **1.348** | **1.187** | **0.989** | 5.541 | 5.601 | 0.831 | 4.416 | 4.438 | 0.878 | 1.369 | 1.252 | 0.988 |
| Pgp (occluded) | 1.727 | 1.677 | **0.979** | 2.875 | 2.847 | 0.945 | 1.916 | 1.882 | 0.974 | **1.714** | **1.673** | 0.979 |
| Pgp (collapsed) | **1.309** | **1.261** | **0.988** | 3.497 | 3.482 | 0.914 | 1.781 | 1.754 | 0.977 | 1.453 | 1.424 | 0.984 |
| Pma1 (active) | **2.046** | **1.776** | **0.973** | 3.204 | 2.935 | 0.927 | 3.228 | 2.938 | 0.930 | 2.164 | 1.907 | 0.967 |
| Pma1 (inhibited) | **1.999** | **1.590** | **0.979** | 7.444 | 7.156 | 0.745 | 7.299 | 6.974 | 0.751 | 2.223 | 1.829 | 0.971 |
| CYP (open) | 4.167 | 3.946 | 0.957 | 11.971 | 11.868 | 0.635 | 8.342 | 8.242 | 0.804 | **3.918** | **3.720** | **0.959** |
| CYP (closed) | **2.004** | **1.552** | **0.990** | 13.395 | 13.262 | 0.594 | 6.132 | 5.986 | 0.857 | 2.089 | 1.763 | 0.987 |
| YbbAP (bound) | **1.320** | **0.678** | **0.997** | 3.586 | 3.273 | 0.931 | 3.625 | 3.360 | 0.926 | 1.695 | 1.161 | 0.991 |
| YbbAP (unbound) | **2.454** | **2.039** | **0.974** | 5.254 | 4.992 | 0.865 | 7.069 | 6.825 | 0.823 | 2.762 | 2.400 | 0.964 |

Table B2: **Evaluation of samples chosen by map-model fit.** We report the Root Mean Square Deviation (RMSD) for all atoms, the Cα RMSD, the template-modeling (TM) score, and the real-space correlation coefficient (RSCC). For CH67, we also report the RMSD for the Cα atoms in the CDR H3 loop (local RMSD). The mean and 95% confidence interval across 3 replicates are reported for the best of 25 samples as assessed by map fit (highest RSCC). **Bold** indicates (statistically significant) best value(s).

| *Structure* | *Metrics* | **CryoBoltz** | **Boltz-1** | **AlphaFold3** |
|---|---|---|---|---|
| STP10 (inward) | RMSD (Å, ↓) | **1.1756** ± **0.0859** | 3.8313 ± 0.0810 | 1.3250 ± 0.0216 |
| | Cα RMSD (Å, ↓) | **0.4776** ± **0.0262** | 3.5838 ± 0.0913 | 0.6543 ± 0.0248 |
| | TM score (↑) | **0.9969** ± **0.0002** | 0.8620 ± 0.0043 | 0.9939 ± 0.0003 |
| | RSCC (↑) | **0.8525** ± **0.0043** | 0.2402 ± 0.0031 | 0.7343 ± 0.0029 |
| STP10 (outward) | RMSD | **0.9090** ± **0.0308** | 2.6596 ± 0.0039 | 4.4777 ± 0.0244 |
| | Cα RMSD | **0.4670** ± **0.0613** | 2.4203 ± 0.0102 | 4.2283 ± 0.0369 |
| | TM score | **0.9971** ± **0.0006** | 0.9485 ± 0.0004 | 0.8276 ± 0.0022 |
| | RSCC | **0.8862** ± **0.0064** | 0.5098 ± 0.0015 | 0.1946 ± 0.0011 |
| CH67 antibody | RMSD | **1.2957** ± **0.2995** | 1.9940 ± 0.0810 | 1.9719 ± 0.1210 |
| | Cα RMSD | **0.8739** ± **0.2010** | 1.4970 ± 0.0418 | 1.4935 ± 0.0902 |
| | Local RMSD | **1.3326** ± **1.0667** | 3.5518 ± 0.1133 | 3.9562 ± 0.4517 |
| | TM score | **0.9917** ± **0.0027** | 0.9713 ± 0.0029 | 0.9708 ± 0.0035 |
| | RSCC | **0.9521** ± **0.0041** | 0.8452 ± 0.0155 | 0.8466 ± 0.0117 |
| P-glycoprotein (apo) | RMSD | **1.4931** ± **0.0803** | 7.0928 ± 0.1115 | 3.8272 ± 1.4344 |
| | Cα RMSD | **1.3377** ± **0.1124** | 7.3054 ± 0.1164 | 3.8652 ± 1.5225 |
| | TM score | **0.9876** ± **0.0004** | 0.7650 ± 0.0054 | 0.9036 ± 0.0611 |
| | RSCC | **0.8157** ± **0.0007** | 0.4261 ± 0.0033 | 0.5797 ± 0.0982 |
| P-glycoprotein (inward) | RMSD | **1.4311** ± **0.0346** | 5.6799 ± 0.1275 | 2.6923 ± 1.1995 |
| | Cα RMSD | **1.2828** ± **0.0407** | 5.7454 ± 0.1302 | 2.6634 ± 1.2533 |
| | TM score | 0.9876 ± 0.0007 | 0.8293 ± 0.0056 | **0.9475** ± **0.0440** |
| | RSCC | **0.8165** ± **0.0011** | 0.5109 ± 0.0069 | 0.6927 ± 0.0884 |

| | | | | |
|---|---|---|---|---|
| P-glycoprotein (occluded) | RMSD | **1.8070** ± **0.0094** | 2.9844 ± 0.1159 | 3.4641 ± 0.0747 |
| | Cα RMSD | **1.7505** ± **0.0080** | 2.9566 ± 0.1160 | 3.4429 ± 0.0738 |
| | TM score | **0.9778** ± **0.0004** | 0.9402 ± 0.0041 | 0.9209 ± 0.0021 |
| | RSCC | **0.7662** ± **0.0006** | 0.6191 ± 0.0072 | 0.5857 ± 0.0001 |
| P-glycoprotein (collapsed) | RMSD | **1.3881** ± **0.0349** | 3.4253 ± 0.2934 | 4.5681 ± 0.1824 |
| | Cα RMSD | **1.3397** ± **0.0363** | 3.4117 ± 0.2929 | 4.5538 ± 0.1798 |
| | TM score | **0.9860** ± **0.0007** | 0.9174 ± 0.0128 | 0.8642 ± 0.0084 |
| | RSCC | **0.7884** ± **0.0012** | 0.5746 ± 0.0330 | 0.4904 ± 0.0108 |
| Pma1 (active) | RMSD | **2.1745** ± **0.3212** | 3.0098 ± 0.2575 | 6.6283 ± 1.0733 |
| | Cα RMSD | **1.9144** ± **0.3517** | 2.7817 ± 0.2590 | 6.3890 ± 1.0764 |
| | TM score | **0.9706** ± **0.0082** | 0.9352 ± 0.0114 | 0.7691 ± 0.0523 |
| | RSCC | **0.5633** ± **0.0113** | 0.3540 ± 0.0234 | 0.2054 ± 0.0102 |
| Pma1 (inhibited) | RMSD | **2.0499** ± **0.1296** | 6.1404 ± 0.7005 | 8.2569 ± 0.5666 |
| | Cα RMSD | **1.6640** ± **0.1413** | 5.8287 ± 0.7180 | 8.0042 ± 0.5289 |
| | TM score | **0.9774** ± **0.0026** | 0.7943 ± 0.0339 | 0.7167 ± 0.0202 |
| | RSCC | **0.5276** ± **0.0012** | 0.2439 ± 0.0246 | 0.2166 ± 0.0027 |
| CYP201A1 (open) | RMSD | **4.3939** ± **0.0988** | 8.5324 ± 0.1545 | 6.5074 ± 0.0836 |
| | Cα RMSD | **4.1473** ± **0.0923** | 8.4389 ± 0.1539 | 6.3770 ± 0.0791 |
| | TM score | **0.9535** ± **0.0013** | 0.7878 ± 0.0110 | 0.8912 ± 0.0036 |
| | RSCC | **0.7693** ± **0.0013** | 0.5074 ± 0.0059 | 0.6220 ± 0.0043 |
| CYP201A1 (closed) | RMSD | **2.8636** ± **0.6566** | 8.7837 ± 0.3684 | **3.5848** ± **0.5770** |
| | Cα RMSD | **2.4669** ± **0.7572** | 8.6674 ± 0.3758 | **3.3910** ± **0.6050** |
| | TM score | **0.9868** ± **0.0022** | 0.7435 ± 0.0151 | 0.9459 ± 0.0186 |
| | RSCC | **0.8062** ± **0.0002** | 0.4937 ± 0.0059 | 0.6830 ± 0.0338 |
| YbbAP (bound) | RMSD | **1.3842** ± **0.0803** | 3.6264 ± 0.2116 | 3.7488 ± 0.1888 |
| | Cα RMSD | **0.7578** ± **0.0892** | 3.3550 ± 0.2512 | 3.4802 ± 0.1897 |
| | TM score | **0.9961** ± **0.0006** | 0.9283 ± 0.0092 | 0.9219 ± 0.0079 |
| | RSCC | **0.7046** ± **0.0005** | 0.5074 ± 0.0116 | 0.4966 ± 0.0083 |
| YbbAP (unbound) | RMSD | **3.8623** ± **0.6404** | 8.2798 ± 0.2012 | **4.0646** ± **0.4578** |
| | Cα RMSD | **3.5271** ± **0.7067** | 8.0973 ± 0.2010 | **3.7913** ± **0.4791** |
| | TM score | **0.9607** ± **0.0058** | 0.7751 ± 0.0066 | 0.9114 ± 0.0184 |
| | RSCC | **0.7098** ± **0.0011** | 0.4944 ± 0.0111 | 0.5594 ± 0.0178 |

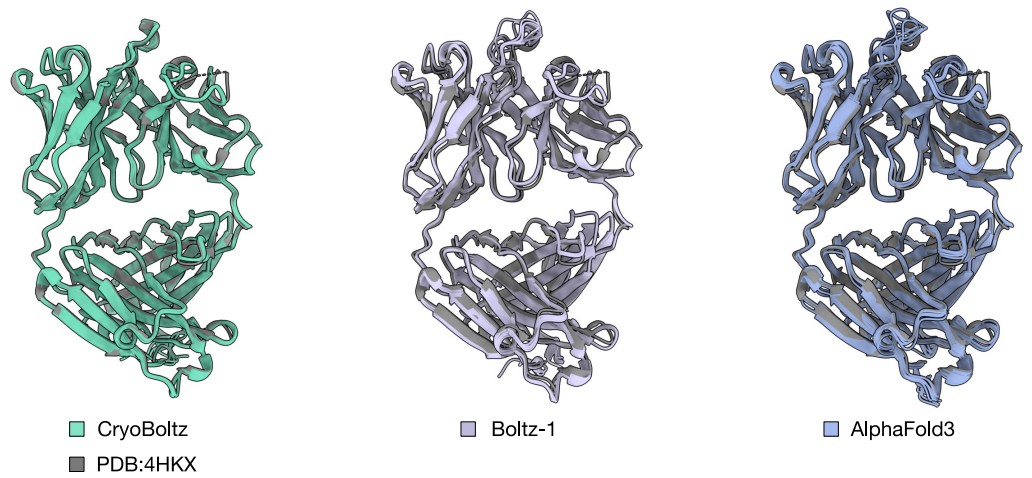

Figure B1: **Full CH67 antibody [70] structures generated by all methods.** Top 5 samples from each method, as ranked by all-atom RMSD.

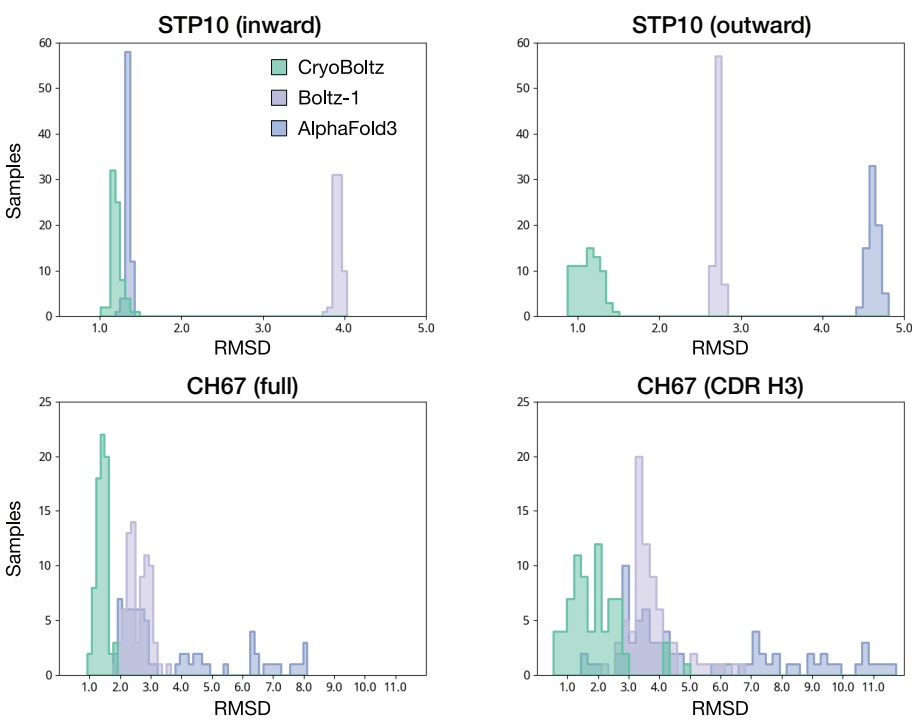

Figure B2: **Distribution of RMSD values for synthetic maps.** All-atom RMSD values for all samples over all replicates are visualized as histograms with 50 bins.

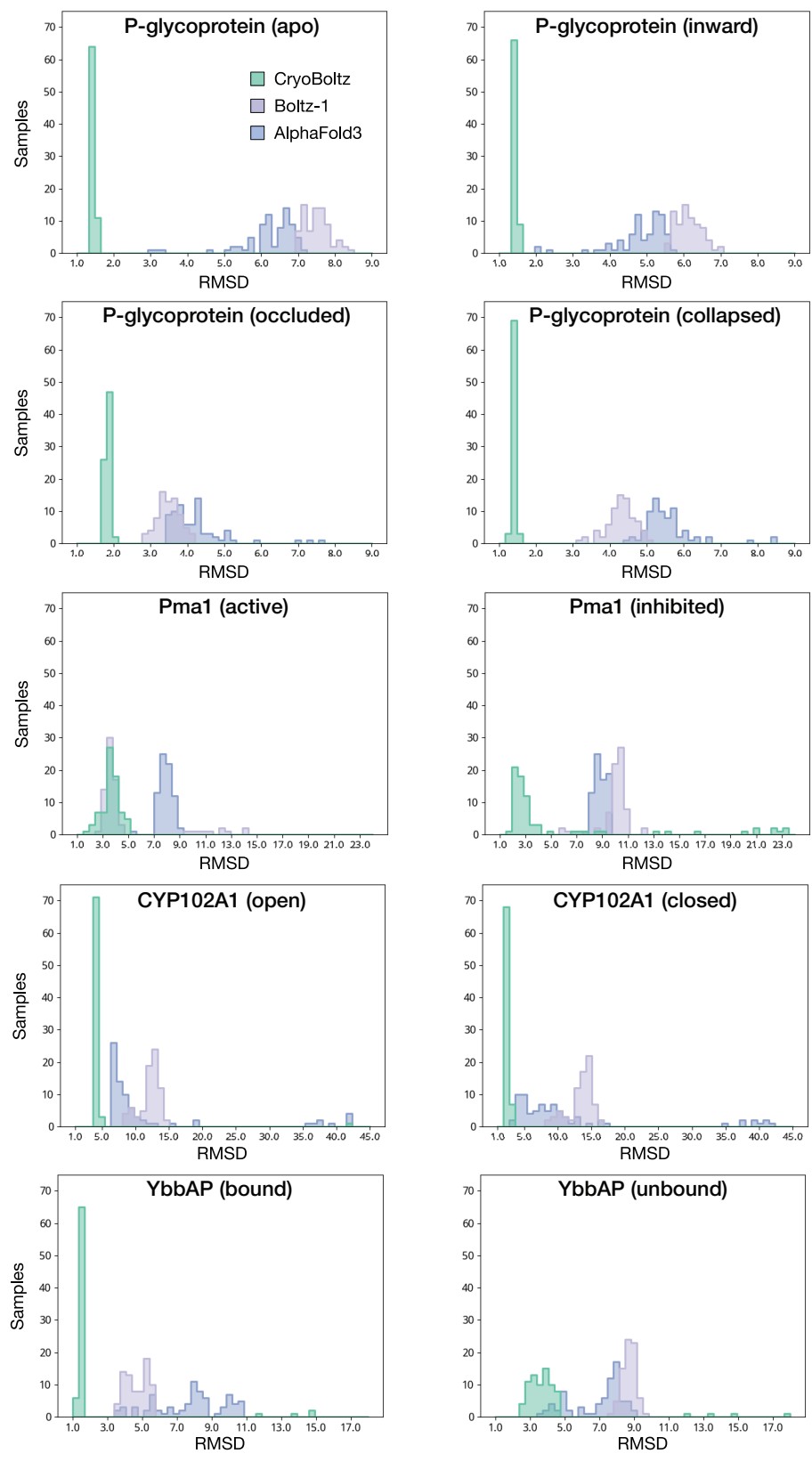

Figure B3: **Distribution of RMSD values for experimental maps.** All-atom RMSD values for all samples over all replicates are visualized as histograms with 50 bins.

