# OpenReview forum: "Multiscale guidance of protein structure prediction with heterogeneous cryo-EM data"
_NeurIPS.cc/2025/Conference — NeurIPS 2025 poster_

### Official Review · Reviewer_jQC6 · 2025-06-24

**Clarity:** 4
**Significance:** 3
**Originality:** 2
**Rating:** 4
**Confidence:** 4

**Summary:**

This paper proposes a guided diffusion procedure for sampling from AlphaFold3 while conditioning on density maps obtained from Cryo-EM experiments. The authors propose a two-stage, global-to-local sampling scheme. In the global stage, the diffusion model is guided towards the course structure of the protein conditioning on sequence alone, and in the local stage, Cryo-EM information is incorporated through the likelihood of the forward model. Since the required guidance term is intractable, a previously proposed approximation to it is used.

The authors evaluate their method on several protein structures, including some simulated and some experimental Cryo-EM maps.

**Questions:**

-	Have the authors considered the possibility of test set contamination? I.e., to what extent are the experimental results attributable to model generalization vs. memorization of structures in the train set? One standard way to measure this would be to measure the sequence (or structure) similarity of each of the test proteins to the nearest sequence/structures in the AF3 train set.
-	Could the authors explain why ModelAngelo performs so poorly in Figure 5? Is it simply because it cannot handle proteins with multiple states well?

**Ethical Concerns:**

["NO or VERY MINOR ethics concerns only"]

**Final Justification:**

As I still have concerns about the degree of evidence the authors have presented that their method will generalize, I hold my score.

**Limitations:**

-	The number of proteins demonstrated is very small – only three. It is hard to avoid the possibility of cherry-picking or a lucky selection. In that sense I think the experimental section is lacking and not statistically significant.
-	As mentioned above, I do not see any effort to measure the possibility of test-set contamination.
-	Two of the three systems involve synthetic rather than experimental Cryo-EM maps. I would like to see more actual Cryo-EM maps used. Actual maps are noisy and potentially pose a much more difficult problem.

**Quality:**

4

**Strengths And Weaknesses:**

Quality

The quality of the paper is good. It is well written and figures are well-presented.

Clarity

The clarity of the exposition is above average, and I enjoyed reading the paper.

Significance

I would judge the significance of this work as moderate-to-high. The field being tackled is extremely active, and many researchers in structural biology are targeting this problem. As time goes on, Cryo-EM microscopy will become more and more common. A lack of multi-conformational accuracy is one of the few remaining gaps in AF3’s performance, and this model appears to do a decent job of addressing that.

Originality

Originality seems moderate. I do believe that this idea was “in the air” somewhat, in that I expect other groups to publish similar work. Guidance of diffusion models for proteins is not new, and it’s only a matter of time before it was successfully applied to Cryo-EM. However, the exact implementation, going from global to local, seems like an original advance, and appears to be very important for getting the method to work well in practice.

---

> ### Author Rebuttal · Authors · 2025-07-31
>
> We thank the reviewer for their thoughtful comments on our work. We are glad the reviewer appreciates the impact of this work in multi-conformation structure analysis, the strength of the multi-scale guidance strategy, and the clarity of the writing. As the reviewer highlights, atomic model building into cryo-EM maps is an active research area which stands to benefit from the recent advances in biomolecular structure prediction, especially to overcome the limitations of manual model building and low resolution maps. Through several additional experiments on real data, as described below, we hope to showcase the applicability of our approach for broadening the scope of cryo-EM map interpretation.
>
> **Additional Validation on Experimental Data.**
> We thank the reviewer for the suggestion to further evaluate our approach on experimental data. We assess our method on 3 additional protein systems associated with real cryo-EM density maps, and whose deposited models are not part of the Boltz-1 training set. For each experimental map, we obtain 25 samples from each method, choose the best sample as assessed by map-model fit (real-space correlation coefficient), and report its all-atom RMSD and TM-score with respect to the deposited model. For ModelAngelo, we obtain its single prediction and report model completeness and TM-score. Metrics are presented in the table below, and we summarize the results here:
> - *Pma1*: We use two experimental maps of this ATPase [EMD-64135, 64136], corresponding to its active and inhibited states [PDB: 9UGB, 9UGC]. Unguided Boltz only samples the active state, whereas guidance also samples the inhibited state with an RMSD of 2.01 Å. AlphaFold3 predictions are not accurate with respect to either state.
> - *CYP102A1*: We use two experimental maps of this oxygenase [EMD-27534, 27536], of resolutions 4.4 Å and 6.5 Å, corresponding to its open and closed states [PDB: 8DME, 8DMG]. Guidance improves the RMSD of the closed state from 7.89 Å to 2.39 Å over unguided Boltz-1, whereas for the lower resolution open state map, a more modest improvement from 7.93 Å to 4.33 Å is observed. ModelAngelo only models 19% of the 4.4 Å map and none of the 6.5 Å map.
> - *YbbAP*: We use two experimental maps of this transporter [EMD-51292, 51291], corresponding to states in which ATP is bound or unbound [PDB: 9GE7, 9GE6]. Unguided Boltz only samples the bound state, which guidance additionally improves to an RMSD of 1.31 Å. The unbound state is also obtained through guidance. Since the unbound state map is of medium resolution at 4.05 Å, ModelAngelo only models 55% of the structure.
>
> Our results on these systems, along with the P-glycoprotein example in the manuscript (Figure 5), demonstrate our method’s ability to recover diverse conformations from real experimental maps. We will include these new results, including visualizations, in our revised manuscript.
>
> |*Method*   |*Metrics*|Pma1 (active)|Pma1 (inhibited)|CYP (open)|CYP (closed)|YbbAP (bound)|YbbAP (unbound)|
> |---------  |---------|:-----------:|:--------------:|:--------:|:----------:|:-----------:|:-----------:|
> |           |Res. (Å) |3.25         |3.52            |6.50       |4.40         |3.66         |4.05         |
> |CryoBoltz  |RMSD     |**1.869**    |**2.010**       |**4.326** |**2.387**   |**1.312**    |**3.087**    |
> |           |TM       |**0.979**    |**0.978**       |**0.956** |**0.988**   |**0.996**    |**0.965**    |
> |Boltz-1    |RMSD     |2.627        |7.559           |7.934     |7.888       |3.547        |8.819        |
> |           |TM       |0.953        |0.740           |0.823     |0.780       |0.930        |0.761        |
> |AlphaFold3 |RMSD     |7.177        |8.493           |6.447     |3.147       |3.613        |3.882        |
> |           |TM       |0.744        |0.711           |0.889     |0.960       |0.927        |0.919        |
> |ModelAngelo|Comp.    |91.47        |72.81           |0.00      |18.91       |81.69        |55.41        |
> |           |TM       |0.889        |0.721           |0.000     |0.102       |0.801        |0.548        |
>
> **ModelAngelo Performance.**
> ModelAngelo is trained solely on cryo-EM maps of resolution 4 Å or better. This accounts for its poor performance on lower resolution maps, especially those of P-glycoprotein and CYP102A1. In contrast, our method’s use of a powerful generative prior aids in fitting atomic structure into lower resolution density.
>
> **Memorization vs Generalization.**
> We thank the reviewer for suggesting checks of test set contamination. We compute the highest structure match score between the tested deposited models and PDB entries contained in the Boltz-1 training set. The structures of the CYP102A1 and YbbAP systems have no matching entries. One state of Pma1 has a match score of 20%, while the other has no matching entries. Finally, the match scores of 3 of the 4 P-Glycoprotein states (Figure 5) are between 40% and 48%, while the last state has no matching entries. Therefore, most structures have little to no overlap with the training dataset, validating that the generalization capabilities of structure prediction models extend to the case of inference-time guidance. We further emphasize that even in cases of structural overlap, as in P-glycoprotein, unguided Boltz-1 fails to sample multiple states, making guidance a useful intervention.

---

> > ### Comment · Reviewer_jQC6 · 2025-08-05
> >
> > Thank you for your response, and for the new experimental results, they partially address my concerns. Adding three additional systems is good, but that leads to a total of six systems, which is still difficult to assess as significant. Can the authors please explain how these six were chosen? I need some confidence that they weren't cherry-picked and that I can expect the results to generalize.
> >
> > As for the similarity, I appreciate the structural similarity computation. Was this done via FoldSeek? Could the authors please also perform a sequence similarity computation using, e.g. MMSeqs2? This practice is quite standard when defining train/test splits in the protein structure prediction field.
> >
> > As I still have concerns about the degree of evidence the authors have presented that their method will generalize, I hold my score.

---

> > > ### Author Response · Authors · 2025-08-07
> > >
> > > We thank the reviewer for taking the time to understand our work. In selecting the experimental datasets, our goal was to curate lower resolution maps of dynamic complexes that satisfy the following criteria:
> > > - Associated with 2+ density maps representing different conformational states
> > > - Resolution of 4 Å or worse for at least one of the maps
> > > - The corresponding deposited atomic models are not within the Boltz-1 training set
> > > - Composed of 2 or fewer unique chains, to obtain accurate unguided predictions from Boltz-1
> > > - Fewer than ~2,200 residues in total, which we find to be the upper memory limit of Boltz-1
> > >
> > > We will describe our dataset curation in our revised supplementary materials.
> > >
> > > The structure match scores we report in our previous rebuttal are computed with the RCSB PDB’s _Structure Similarity_ search tool, which uses BioZernike structure descriptors. The sequence similarities, as computed with MMSeqs2 within the RCSB PDB _Sequence Similarity_ search tool, are 0% for YbbAP chain A, 44% for YbbAP chain B, 78% for Pma1, 97% for CYP102A1, and 100% for P-Glycoprotein. While some of these sequences share significant identity with the training dataset, Boltz-1 fails to sample the experimentally observed conformations of these systems, so we believe the structure match scores are a more informative measure.
> > >
> > > For the synthetic data examples (Figures 3 and 4), the deposited structures are within the Boltz-1 training set, but the model is still unable to sample correct conformations in the absence of guidance. These synthetic experiments therefore serve to validate the guidance approach.
> > >
> > > While practical considerations prevent a more comprehensive sweep, we hope to do so with synthetic data in future work, and believe the 13 (real and synthetic) maps evaluated in this study demonstrate the applicability of our method. We additionally envision that the flexibility of our inference-time guidance approach will benefit from the parallel developments of more powerful structure predictors.

---

> > > > ### Comment · Reviewer_jQC6 · 2025-08-08
> > > >
> > > > >In selecting the experimental datasets, our goal was to curate lower resolution maps of dynamic complexes that satisfy the following criteria:
> > > >
> > > > Thanks for the clarification. The result of these filters was only six systems? And your model succeeds on all six? The fact that Boltz is such a recent model does make evaluation difficult here since that excludes a lot from the training set. 6/6 successes is nice but still very worryingly few.
> > > >
> > > > The high sequence similarity cases do worry me, especially since they comprise almost half of your training set. Could you try running something like MSA subsampling for AlphaFold2 for these cases? E.g. https://www.nature.com/articles/s41586-023-06832-9

---

> > > > > ### Author Response · Authors · 2025-08-09
> > > > >
> > > > > We chose the 4 experimental systems as representative examples by manually searching the EMDB and cross-referencing the PDB. We agree that systematically building a larger benchmark would add to the validation of our method. We also thank the reviewer for the suggestion of MSA subsampling and can include this as a baseline in our revised manuscript.

---

### Official Review · Reviewer_XC5a · 2025-06-30

**Clarity:** 3
**Significance:** 3
**Originality:** 3
**Rating:** 4
**Confidence:** 4

**Summary:**

This paper presents CryoBoltz, a method that bridges protein structure prediction models like AlphaFold3 with cryo-EM data via multiscale guidance, driving sampling towards conformational states consistent with experimental observations. By integrating global and local structural constraints from density maps, CryoBoltz overcomes the single-structure bias of existing models, enabling accurate atomic modeling of heterogeneous cryo-EM maps for dynamic systems like transporters and antibodies.

**Questions:**

- What if AlphaFold3 fails to accurately predict the protein structure? I suppose that if the AF-predicted structures can fit the density up to some rotation and translation of some part of the structure, the error can be corrected. But if there are some larger errors, can the density correct the AF-3 structures?
- Is it possible to incorporate the cryo-EM data into the training process, like CryoSTAR? Maybe adding a learable pair representation at the template module to finetune the Boltz-1 is enough.
- I think incorporating some validation metrics (to avoid the prior from AF3)  will potentially boost the effectiveness of this study.

**Ethical Concerns:**

["NO or VERY MINOR ethics concerns only"]

**Final Justification:**

The authors answered my questions and I have no further concerns. I think this paper bridge the gap between cryoem and protein structure prediciton with a smart approach.

**Quality:**

3

**Strengths And Weaknesses:**

Strengths

- Utilizes AlphaFold3 as a prior to enhance the accuracy of heterogeneous analysis, effectively integrating rich sequence and biophysical priors with cryo-EM data to drive predictions toward experimental consistency.
- Implements a smart design of global and local guidance mechanisms: global guidance targets large-scale conformational adjustments using low-resolution point cloud representations, while local guidance refines high-resolution details based on physics-based forward models.



Weaknesses

- There is a vulnerability in relying on AlphaFold3 as a prior. If AlphaFold3 fails to accurately predict the protein structure, the entire heterogeniety analysis accuracy relying on it may be severely compromised, potentially leading to incorrect interpretations of the data.
- The potential biases introduced by AlphaFold3 pose an issue. For instance, if it mispredicts a loop region as a helix while getting the global structure correct, it's unclear whether the cryo - EM data can effectively correct such local mispredictions. There is also a lack of clarity on how to detect such biases and whether there are failure cases presented to alert users and prevent them from being misled by the model.

---

> ### Author Rebuttal · Authors · 2025-07-31
>
> We thank the reviewer for their thoughtful comments on our work. We are glad the reviewer appreciates the use of AlphaFold3 as a strong biophysical prior for atomic modeling into cryo-EM maps, as well as the novelty of the proposed multiscale guidance strategy. While we show through our experiments that our approach broadens the scope of cryo-EM model building, we also hope that our flexible inference-time method will benefit from the parallel advances in biomolecular structure prediction. We elaborate on these aspects below.
>
> **Robustness of the Prior.**
> The reviewer is correct that our method inherits both the strengths and limitations of the underlying structure prediction model. We aim to take advantage of the strong biophysical priors learned by AlphaFold3/Boltz-1, which is beneficial especially when fitting atomic models to lower resolution maps (Figure 5). Conversely, for complexes that are badly mispredicted by the base model, including full antibody complexes and DSL1 which we note in the Discussion section, guidance is unable to overcome this bias. However, a key strength of our inference-time approach is that it can be applied to any structure prediction model with a diffusion-based sampler. We thus envision that the continuous developments in structure prediction will further boost the ability of our guidance strategy to model highly dynamic complexes.
>
> **Validation Metrics.**
> We thank the reviewer for the suggestion to expand the validation metrics. Map-model fit, as assessed by real-space correlation coefficient as in Supplementary Table B1, is one approach to model validation and selection. Additional measures of model quality such as MolProbity score or Q score would indeed serve as important validation, and we will include these in our revised manuscript. We will additionally conduct analyses of Boltz-1 confidence scores, which may serve as another mechanism for a user to detect failure cases.
>
> **Training with Cryo-EM Maps.**
> While the flexibility of inference-time guidance is a strength of our approach, fine-tuning Boltz with map conditioning, potentially incorporating our guidance algorithm, represents an interesting avenue for future work. We thank the reviewer for this suggestion.

---

> > ### Comment · Reviewer_XC5a · 2025-08-01
> >
> > Thank you for your response. I do not have further concerns.

---

### Official Review · Reviewer_Cxm9 · 2025-07-02

**Clarity:** 3
**Significance:** 2
**Originality:** 2
**Rating:** 4
**Confidence:** 2

**Summary:**

The paper propose a method for guiding generative models for protein stuctures (from sequence data) based on observed experimental densities. The guiding is done using two approaches: local (density based) and global (geometry/structure based, derived form the densities). The method builds on the idea of posterior sampling, where the AF3 type models act as priors and the observed densities define the likelihoods. The paper shows that this approach leads to more accurate structures than the unconditional/unguided sampling methodds (AF3 and Boltz-1)

**Questions:**

The validation was done on one structure for which there was real experimental data. To what extend can we believe the method will generalize to other structures?

To what extend are the reduction in RMSD important? The structures obtained by guidance are naturally going to be closer to "ground truth" resolved structures, which are probably resolved using cryoEM data? But the unguided structures may very well be accurate too, it's just that they have not been observed in such states in cryoEM. Now, I may be talking nonsense here as I don't know the field too well, but my point is, I expect a bias of some sort but can't quite put my finger on it. Perhaps the authors could either alleviate my concern, or transparently lay down the limitations of this validation.

**Ethical Concerns:**

["NO or VERY MINOR ethics concerns only"]

**Final Justification:**

Based on the response I moved my score upwards from borderline reject to borderline accept. I think the additional clarifications are useful and validated the significance of the paper. Since this is not my primary field of interest (structural biology) I'm not particularly impressed by the approach taken or its novelty, however, the paper is sound and seems a valuable contribution to DL-based structure prediction. And can be impactful as such.

**Limitations:**

Overall limitations and impact have been adequately addressed. The scientific impact could be more deeply discussed though (cf the questions field)

**Paper Formatting Concerns:**

no concerns

**Quality:**

3

**Strengths And Weaknesses:**

# Strengths

1.	The paper is very well written
2.	The formulation of the problem (posterior inference) is elegant and the derivation is sound.
3.	The results are good.

# Weaknesses

1.	The validation seems a bit biased by design since Table 1 is made using synthetic maps. This means that the maps are rendered from a know structure, so we know the to-be-recovered structure is indeed contained in this rendered density maps. Wouldn’t the guidance necessarily lead to the optimal structures? What kind of noise is used in the rendering and how realistic is it? Should this table then be considered more as a sanity check? I do appreciate that this table at confirms the benefit of including observational data in the generative process.
2.	The validation seems limited considering 2/3 are based on synthetic density maps. The 3rd problem is on experimental density maps. Although the it confirms the applicability on experimental data, the validation may be too limited to draw general conclusions.

# Minor
1.	The paper is applied and no new general principles are presented that generalize outside of the domain of structure prediction. The new guiding term (global) seems a bit ad-hoc but effective. The general principles used are known in the broader context of inverse modeling. However, within the specific application domain the paper seems relevant.
2.	The paper explains the contribution as “multi-scale”. However I do not see why one is called global and the other local. The local is based on pixel-wise errors and in that sense can be considered local. However, all pixels are used, and the signal is in that sense global. The named global term is based on based distances between point clouds.  Wouldn’t a more informative naming be density vs geometry guidance or something? The main difference is imo not local vs global, but the fact that the “global” term directly gives a supervisory signal on the actual geometry. E.g. a small shift of the structure could lead to a big change in density error, whereas a small shift remains a small distance in the geometric loss. Since the problem is geometric, the “global” terms seems to be the most suitable guiding term, and I don’t understand why we would need the local term at all. In fact, the experiments show that global term is the superior one.

---

> ### Author Rebuttal · Authors · 2025-07-31
>
> We thank the reviewer for their thoughtful comments on our work. We are glad the reviewer appreciates the clarity of the writing and the strength of the results. Our primary motivation is to fit atomic structures into cryo-EM maps of dynamic complexes, which often requires time-consuming manual effort especially on lower resolution maps where current automated methods tend to fail. By leveraging the powerful generative prior of AlphaFold3, our method provides a new framework for cryo-EM model building that aims to overcome these limitations. We present several additional experiments on real data to further showcase the general applicability of our approach.
>
> **Additional Validation on Experimental Data.**
> We thank the reviewer for the suggestion to further evaluate our approach on experimental data. We assess our method on 3 additional protein systems associated with real cryo-EM density maps, and whose deposited models are not part of the Boltz-1 training set. For each experimental map, we obtain 25 samples from each method, choose the best sample as assessed by map-model fit (real-space correlation coefficient), and report its all-atom RMSD and TM-score with respect to the deposited model. For ModelAngelo, we obtain its single prediction and report model completeness and TM-score. Metrics are presented in the table below, and we summarize the results here:
> - *Pma1*: We use two experimental maps of this ATPase [EMD-64135, 64136], corresponding to its active and inhibited states [PDB: 9UGB, 9UGC]. Unguided Boltz only samples the active state, whereas guidance also samples the inhibited state with an RMSD of 2.01 Å. AlphaFold3 predictions are not accurate with respect to either state.
> - *CYP102A1*: We use two experimental maps of this oxygenase [EMD-27534, 27536], of resolutions 4.4 Å and 6.5 Å, corresponding to its open and closed states [PDB: 8DME, 8DMG]. Guidance improves the RMSD of the closed state from 7.89 Å to 2.39 Å over unguided Boltz-1, whereas for the lower resolution open state map, a more modest improvement from 7.93 Å to 4.33 Å is observed. ModelAngelo only models 19% of the 4.4 Å map and none of the 6.5 Å map.
> - *YbbAP*: We use two experimental maps of this transporter [EMD-51292, 51291], corresponding to states in which ATP is bound or unbound [PDB: 9GE7, 9GE6]. Unguided Boltz only samples the bound state, which guidance additionally improves to an RMSD of 1.31 Å. The unbound state is also obtained through guidance. Since the unbound state map is of medium resolution at 4.05 Å, ModelAngelo only models 55% of the structure.
>
> Our results on these systems, along with the P-glycoprotein example in the manuscript (Figure 5), demonstrate our method’s ability to recover diverse conformations from real experimental maps, especially at resolutions where other methods fail. We will include these results, including visualizations, in our revised manuscript.
>
> |*Method*   |*Metrics*|Pma1 (active)|Pma1 (inhibited)|CYP (open)|CYP (closed)|YbbAP (bound)|YbbAP (unbound)|
> |---------  |---------|:-----------:|:--------------:|:--------:|:----------:|:-----------:|:-----------:|
> |           |Res. (Å) |3.25         |3.52            |6.50       |4.40         |3.66         |4.05         |
> |CryoBoltz  |RMSD     |**1.869**    |**2.010**       |**4.326** |**2.387**   |**1.312**    |**3.087**    |
> |           |TM       |**0.979**    |**0.978**       |**0.956** |**0.988**   |**0.996**    |**0.965**    |
> |Boltz-1    |RMSD     |2.627        |7.559           |7.934     |7.888       |3.547        |8.819        |
> |           |TM       |0.953        |0.740           |0.823     |0.780       |0.930        |0.761        |
> |AlphaFold3 |RMSD     |7.177        |8.493           |6.447     |3.147       |3.613        |3.882        |
> |           |TM       |0.744        |0.711           |0.889     |0.960       |0.927        |0.919        |
> |ModelAngelo|Comp.    |91.47        |72.81           |0.00      |18.91       |81.69        |55.41        |
> |           |TM       |0.889        |0.721           |0.000     |0.102       |0.801        |0.548        |
>
> **Validation by RMSD.**
> We agree with the reviewer that samples from the unguided model may also correspond to valid states of a protein. However, our goal is to specifically obtain atomic structures that are consistent with cryo-EM data. Our motivation in baseline comparisons against unguided Boltz-1 and AlphaFold3 is not to suggest that these structures are necessarily incorrect, just that they do not correspond to the target cryo-EM state in the absence of guidance. Guidance provides a mechanism through which the powerful predictive capabilities of these pretrained models can be directed towards the states a cryo-EM practitioner desires from their data.
>
> **Validation on Synthetic Data.**
> While a noiseless synthetic map does correspond perfectly to its target structure, it is not certain a priori that Boltz-1 will be able to sample this structure. Indeed, Boltz-1 never samples the inward conformation of the STP10 system (Section 4.2). As the reviewer suggests, our synthetic experiments can thus be seen as validation that our proposed guidance strategy is effective at driving the sampling process toward a structure consistent with a map.
>
> **Multiscale Guidance.**
> We apologize for the lack of clarity surrounding nomenclature of the multiscale guidance strategy. The terms “global” and “local” are meant to describe the scale of the conformational changes driven by these two guidance phases. The global phase is most effective at guiding coarser domains of the structure, since the Wasserstein distance metric permits a strong gradient even between a sample atom and map point that are far apart.  However, since the point cloud is only a rough approximation of the map, the density errors applied during the local phase are more adept at guiding loops and secondary structure elements into the right locations. While the global guidance phase is indeed responsible for the majority of reduction in RMSD, we find that the local guidance phase remains important for accurate model fitting. Conducting an ablation on the inhibited state of Pma1, global guidance alone reduces RMSD from 7.6 Å to 3.0 Å, while the addition of local guidance further improves accuracy to 2.0 Å. We will include ablations on real data in our revised manuscript.
>
> **Novelty.**
> We agree with the reviewer that the novelty of the method primarily lies within this application domain. That said, we believe our multiscale guidance strategy may have broader relevance and hope that it will inspire other diffusion guidance approaches in 3D small molecule generation or computer vision.

---

### Official Review · Reviewer_SWNS · 2025-07-04

**Clarity:** 3
**Significance:** 3
**Originality:** 2
**Rating:** 5
**Confidence:** 2

**Summary:**

CryoBoltz is an inference-time framework that enhances diffusion-based protein structure predictors by integrating information form heterogeneous cryo-electron. It adds data-driven guidance to a pretrained score model without retraining. Using a multiscale strategy over four stages it drives models toward conformations consistent with experimental data. The stages include initial unguided sampling, coarse global alignment to a point-cloud representation of the density map, fine local fitting against the full map, and final unguided relaxation. CryoBoltz works atop any diffusion-based predictor namely Boltz-1, and overcomes their static bias. Evaluations on synthetic maps for a transporter and an antibody and on real cryo-EM maps of P-glycoprotein demonstrate its ability to recover diverse states with high accuracy. This approach bridges sequence-based models with experimental observables.

**Questions:**

•	Could the authors provide more quantitative comparisons to alternative methods, namely ADP-3D?

•	I appreciate the synthetic dataset generation for th STP10 and CH67 antibodies. Could you show for one of them how the
accuracy of your model changes with the resolution of your synthetic density maps. It would be interesting to see a line plot of resolution vs rmsd for all the compared methods. It would be nice to understand on a single molecule where the method works best and potential where it fails with respect to the other methods?

•	Could the authors test the method on a new pdb structure that is not included in the Boltz-1 training dataset? Is this an area where your inference-time methods shine even more?

•	Could the authors compare their approach on a larger experimental dataset. Its currently hard to judge whether the 4 proteins (section 4.4) are cherry picked or whether the outperformance generalises.

•	Low priority: Does using Boltz-2 further improve your method?

**Ethical Concerns:**

["NO or VERY MINOR ethics concerns only"]

**Final Justification:**

The authors have included three substantial new experiments in the rebuttal phase, including ablations with respect to resolution in their synthetic datasets and further comparison to experiments. These have significantly strengthened the evaluation of their approach and I am therefore increasing my score to accept. The experimental comparisons in particular demonstrate the potential impact of the work presented in the paper. However, as the authors compare against only four experimental structures, I retain some concerns regarding the general applicability of the method. Additionally, the paper is focused on a very specific sub-area of AI. Due to these two points I have not raised my score further.

**Limitations:**

A more detailed experimental section would help in understanding the true limitations of the method.

**Quality:**

3

**Strengths And Weaknesses:**

-	The authors clearly outline their proposed method.
-	The figures are instructive, clear and of high quality.
-	The authors show their multi step guidance is required through various ablations
-	The authors illustrate a failure of the models to reproduce cryo-EM data without any guidance
-	Overall, the complexity of the approach is minimal. Comparison to other approaches and other guidance methods could be extended.
-	The approach of generating synthetic data is interesting but could be leveraged more to investigate the performance as a function of resolution in greater detail.
-	The experimental comparisons are valuable, but the authors only compare on a single protein (P-glycoprotein) with 4 different structures. Could the authors use a larger benchmark dataset to compare their method to show the results aren’t cherry picked?
- Overall the a larger benchmark dataset would be helpful to asses the trust worthiness of the method. If its very hard to obtain cryo-em data you could extend your synthetic generation approach to a relevant subset of the pdb? Or all new PDB structures post Boltz-1 training?
-	The reviewer does not have the domain expertise to judge the significance of this work for the broader bio community.

---

> ### Author Rebuttal · Authors · 2025-07-31
>
> We thank the reviewer for their thoughtful comments on our work. We are glad the reviewer appreciates the clarity of the writing and the value of the synthetic experiments. As the first method to leverage the state-of-the-art AlphaFold3 as a prior for cryo-EM model building, we believe our method will broaden the structural interpretation of experimental data, and hope to showcase its applicability through the additional experiments we detail in our response below.
>
> **Additional Validation on Experimental Data.**
> We thank the reviewer for the suggestion to further evaluate our approach on experimental data. We assessed our method on 3 additional protein systems associated with real cryo-EM density maps, and whose deposited models are not part of the Boltz-1 training set. For each experimental map, we obtain 25 samples from each method, choose the best sample as assessed by map-model fit (real-space correlation coefficient), and report its all-atom RMSD and TM-score with respect to the deposited model. For ModelAngelo, we obtain its single prediction and report model completeness and TM-score. We additionally expand our set of baselines to include ADP-3D, and describe these evaluations further in the **ADP-3D Baseline** section below. Metrics are presented in the table below, and we summarize the results here:
> - *Pma1*: We use two experimental maps of this ATPase [EMD-64135, 64136], corresponding to its active and inhibited states [PDB: 9UGB, 9UGC]. Unguided Boltz only samples the active state, whereas guidance also samples the inhibited state with an RMSD of 2.01 Å. AlphaFold3 predictions are not accurate with respect to either state.
> - *CYP102A1*: We use two experimental maps of this oxygenase [EMD-27534, 27536], of resolutions 4.4 Å and 6.5 Å, corresponding to its open and closed states [PDB: 8DME, 8DMG]. Guidance improves the RMSD of the closed state from 7.89 Å to 2.39 Å over unguided Boltz-1, whereas for the lower resolution open state map, a more modest improvement from 7.93 Å to 4.33 Å is observed. ModelAngelo only models 19% of the 4.4 Å map and none of the 6.5 Å map.
> - *YbbAP*: We use two experimental maps of this transporter [EMD-51292, 51291], corresponding to states in which ATP is bound or unbound [PDB: 9GE7, 9GE6]. Unguided Boltz only samples the bound state, which guidance additionally improves to an RMSD of 1.31 Å. The unbound state is also obtained through guidance. Since the unbound state map is of medium resolution at 4.05 Å, ModelAngelo only models 55% of the structure.
>
> Our results on these systems, along with the P-glycoprotein example in the manuscript (Figure 5), demonstrate our method’s ability to recover diverse conformations from real experimental maps, especially at resolutions where other methods fail. We will include these results, including visualizations, in our revised manuscript.
>
> |*Method*   |*Metrics*|Pma1 (active)|Pma1 (inhibited)|CYP (open)|CYP (closed)|YbbAP (bound)|YbbAP (unbound)|
> |---------  |---------|:-----------:|:--------------:|:--------:|:----------:|:-----------:|:-----------:|
> |           |Res. (Å) |3.25         |3.52            |6.50       |4.40         |3.66         |4.05         |
> |CryoBoltz  |RMSD     |**1.869**    |**2.010**       |**4.326** |**2.387**   |**1.312**    |**3.087**    |
> |           |TM       |**0.979**    |**0.978**       |**0.956** |**0.988**   |**0.996**    |**0.965**    |
> |Boltz-1    |RMSD     |2.627        |7.559           |7.934     |7.888       |3.547        |8.819        |
> |           |TM       |0.953        |0.740           |0.823     |0.780       |0.930        |0.761        |
> |AlphaFold3 |RMSD     |7.177        |8.493           |6.447     |3.147       |3.613        |3.882        |
> |           |TM       |0.744        |0.711           |0.889     |0.960       |0.927        |0.919        |
> |ModelAngelo|Comp.    |91.47        |72.81           |0.00      |18.91       |81.69        |55.41        |
> |           |TM       |0.889        |0.721           |0.000     |0.102       |0.801        |0.548        |
> |ADP-3D     |RMSD     |9.906        |12.119          |-         |-           |34.409       |52.125      |
> |           |TM       |0.934        |0.888           |-         |-           |0.286        |0.091       |
>
> **ADP-3D Baseline.**
> We thank the reviewer for the suggestion to compare ADP-3D, which we evaluate on P-glycoprotein (Figure 5), Pma1, and YbbAP. CYP102A1 produced errors due to its large size. For each cryo-EM map, we provide ADP-3D with the ModelAngelo prediction as the required initial structure, and choose the best of 5 samples by map-model fit. For Pma1 and YbbAP metrics, please see the table in the **Additional Validation on Experimental Data** section above. For P-glycoprotein, we reproduce Table 2 of the manuscript below for the reviewer’s convenience. We find that ADP-3D improves TM-score over ModelAngelo for Pma1 and P-glycoprotein maps, due to some additional modeling of the incomplete initial structure. However, we also observe very poor RMSD scores, especially for YbbAP and P-glycoprotein, where completely incorrect predictions are made for large portions of the structure.
>
> |*Method*   |*Metrics*|Glycoprotein (apo)|Glycoprotein (inward)|Glycoprotein (occluded)|Glycoprotein (collapsed)|
> |---------  |---------|:-----------:|:--------------:|:--------:|:----------:|
> |           |Res. (Å) |4.3    |4.4   |4.1|4.4|
> |CryoBoltz  |RMSD     |**1.381**|**1.329**|**1.745**|**1.340**|
> |           |TM       |**0.989**|**0.989**|**0.979**|**0.986**|
> |Boltz-1    |RMSD     |6.994|5.630|2.929|3.425|
> |           |TM       |0.767|0.828|0.942|0.917|
> |AlphaFold3 |RMSD     |3.827|2.692|3.440|4.568|
> |           |TM       |0.904|0.947|0.921|0.864|
> |ModelAngelo|Comp.    |40.33    |18.25  |2.26|2.53|
> |           |TM       |0.361   |0.134|0.010|0.010|
> |ADP-3D     |RMSD     |27.367|41.998|58.630|62.491|
> |           |TM       |0.583|0.333|0.092|0.073|
>
>
> **Accuracy vs Resolution.**
> We appreciate the reviewer’s suggestion to analyze all-atom RMSD as a function of map resolution. In the table below, we report the mean and standard deviation of 10 samples guided to the inward state of the STP10 system (Figure 3), across varying resolutions of the synthetic map. We find that performance is best at resolutions better than 6 Å. We additionally observe that RMSD slightly increases going from a 2 Å to a 5 Å map, likely due to smoother gradients in the local guidance phase, suggesting that our method may be improved by adopting a frequency marching schedule.
>
> ||||||||||
> |----------------|:---:|:---:|:---:|:---:|:---:|:---:|:---:|:---:|
> |**Res (Å)**     |2.0  |3.0  |4.0  |5.0  |6.0  |8.0  |10.0 |15.0 |
> |**RMSD Mean**   |1.181|1.116|1.064|1.061|1.095|1.276|1.411|1.680|
> |**RMSD Std.**   |0.034|0.048|0.108|0.104|0.102|0.058|0.058|0.058|
>
> **Extending to Boltz-2.**
> We agree with the reviewer that extending our approach to Boltz-2 represents an important area of future work. Since Boltz-2 introduces significant code changes and was released after the NeurIPS deadline, we have not yet integrated our guidance method. The improved predictive capabilities of Boltz-2 may indeed broaden the applicability of our guidance approach.

---

> > ### Comment · Reviewer_SWNS · 2025-08-08
> >
> > I think the added experiments improve the validity of the MS. Specifically because of the extended validation against experiment and also comparisons against existing traditional baselines. For the final manuscript it would be good to extend the testing of the results to a larger experimental dataset. Could you create a benchmark on the entire EMD such that future methods can be objectively compared against your approach/existing methods? How did you choose the 3 specific EMD structures?

---

> > > ### Author Response · Authors · 2025-08-09
> > >
> > > We are glad the reviewer appreciates the additional experimental validation and thank them for their constructive comments.  In selecting the experimental datasets, our goal was to curate lower resolution maps of dynamic complexes that satisfy the following criteria:
> > > - Associated with 2+ density maps representing different conformational states
> > > - Resolution of 4 Å or worse for at least one of the maps
> > > - The corresponding deposited atomic models are not within the Boltz-1 training set
> > > - Composed of 2 or fewer unique chains, to obtain accurate unguided predictions from Boltz-1
> > > - Fewer than ~2,200 residues in total, which we find to be the upper memory limit of Boltz-1
> > >
> > > We will describe our dataset curation in our revised supplementary materials. While practical considerations prevent a more comprehensive sweep, we hope to extend our benchmarking with synthetic data in future work.

---

### Note · Authors · 2025-08-15

We thank the AC and reviewers for engaging with our work. We value that the reviewers appreciate the novelty of the multiscale guidance strategy [XC5a, jQC6], the strength of the results [Cxm9], the significance to multi-conformation structure analysis [jQC6], and the clarity of the writing [SWNS, Cxm9, jQC6]. We believe that our extended experiments during the rebuttal period further highlight our method’s ability to recover diverse conformations from real cryo-EM data, and intend to incorporate all reviewers’ helpful feedback in our revised manuscript.

---

### Decision · Program_Chairs · 2025-09-17

**Decision:**

Accept (poster)

**Comment:**

This paper presents an approach to combing the predictive power of structure prediction models like AlphaFold3 with experimental data from cryoEM in the form of a 3D density.  The approach works by using the density map to guide the structure prediction given the sequence.  The method is demonstrated on three structures, two based on synthetic data and one based on experimental data.  In all cases the result is compared against a ground truth structure and shows significant improvement in predictive accuracy vs baseline methods.

During the rebuttal and discussion phase the authors reported several new experiments on real data, showing further gains and allaying some reviewer concerns.  The final consensus among all reviewers was positive and the AC agrees.  However, the AC strongly suggests the following changes:
 - describe and incorporate the newly reported experimental results
 - incorporate all other feedback from the reviews and discussion
 - modify the title to not explicitly reference AlphaFold3.  The proposed method could, in principle, be used with AlphaFold3 but is never actually demonstrated on it.  An acceptable title could be: "Multiscale guidance of protein structure prediction with heterogeneous cryo-EM data" but I leave the specifics at the authors discretion.